# Discovery of Novel Boron-Containing *N*-Substituted Oseltamivir Derivatives as Anti-Influenza A Virus Agents for Overcoming N1-H274Y Oseltamivir-Resistant

**DOI:** 10.3390/molecules27196426

**Published:** 2022-09-29

**Authors:** Ruifang Jia, Jiwei Zhang, Jian Zhang, Chiara Bertagnin, Anna Bonomini, Laura Guizzo, Zhen Gao, Xiangkai Ji, Zhuo Li, Chuanfeng Liu, Han Ju, Xiuli Ma, Arianna Loregian, Bing Huang, Peng Zhan, Xinyong Liu

**Affiliations:** 1Department of Medicinal Chemistry, Key Laboratory of Chemical Biology (Ministry of Education), School of Pharmaceutical Sciences, Cheeloo College of Medicine, Shandong University, 44 West Culture Road, Jinan 250012, China; 2Institute of Medical Sciences, The Second Hospital, Cheeloo College of Medicine, Shandong University, Jinan 250033, China; 3Department of Molecular Medicine, University of Padova, Via Gabelli 63, 35121 Padova, Italy; 4Institute of Poultry Science, Shandong Academy of Agricultural Sciences, 1 Jiaoxiao Road, Jinan 250023, China; 5China-Belgium Collaborative Research Center for Innovative Antiviral Drugs of Shandong Province, 44 West Culture Road, Jinan 250012, China

**Keywords:** influenza, neuraminidase inhibitors, oseltamivir derivatives, 150-cavity, boronic acid

## Abstract

To address drug resistance to influenza virus neuraminidase inhibitors (NAIs), a series of novel boron-containing *N*-substituted oseltamivir derivatives were designed and synthesized to target the 150-cavity of neuraminidase (NA). In NA inhibitory assays, it was found that most of the new compounds exhibited moderate inhibitory potency against the wild-type NAs. Among them, compound **2c** bearing 4-(3-boronic acid benzyloxy)benzyl group displayed weaker or slightly improved activities against group-1 NAs (H1N1, H5N1, H5N8 and H5N1-H274Y) compared to that of oseltamivir carboxylate (**OSC**). Encouragingly, **2c** showed 4.6 times greater activity than **OSC** toward H5N1-H274Y NA. Moreover, **2c** exerted equivalent or more potent antiviral activities than **OSC** against H1N1, H5N1 and H5N8. Additionally, **2c** demonstrated low cytotoxicity in vitro and no acute toxicity at the dose of 1000 mg/kg in mice. Molecular docking of **2c** was employed to provide a possible explanation for the improved anti-H274Y NA activity, which may be due to the formation of key additional hydrogen bonds with surrounding amino acid residues, such as Arg152, Gln136 and Val149. Taken together, **2c** appeared to be a promising lead compound for further optimization.

## 1. Introduction

Influenza (flu) is an acute respiratory infectious disease that continues to baffle humans by its frequently changing nature, seasonal epidemics and occasional pandemics [1]. Influenza viruses belong to the family of *Orthomyxoviridae* and are categorized as types A, B, C and D, of which only influenza virus A, B and C can infect humans [1,2]. According to the World Health Organization (WHO), influenza affects approximately 1 billion individuals each year resulting in between 290,000 and 650,000 deaths [3,4]. The influenza virus A has caused three major pandemics in the last century, including 1918 Spanish flu (H1N1), 1957 Asian flu (H2N2) and 1968 Hong Kong flu (H3N2) [5]. The 2009 Swine flu (H1N1) caused millions of cases worldwide and had a significant impact on human lives and resources [6,7]. In recent years, highly pathogenic avian influenza viruses of subtype H5N1 and H7N9 exhibited high mortality rates of around 60% and 40%, respectively, resulting in many hospitalizations and deaths [8,9]. Therefore, it is vital to pay more attention to such diseases as a serious and constant threat to global public health [10].

Neuraminidase (NA), a membrane-bound glycoprotein on the viral surface, plays a critical role not only in catalyzing the release of newly formed virions in host cells, but also in facilitating the movement of virions in the respiratory tract [11]. Up to now, four neuraminidase inhibitors (NAIs) were successfully approved to treat influenza virus infections, namely Oseltamivir (Tamiflu) [12], Zanamivir (Relenza) [13], Peramivir (Rapivab) [14] and Laninamivir octanoate (Inavir) (Figure 1) [15]. Among them, oseltamivir is the only agent for oral administration and this NAI has taken over the largest market share. However, with the widespread use of oseltamivir, resistant strains of influenza virus have compromised the therapeutic effects, especially the N1-H274Y mutant [16,17]. Hence, it is urgent to develop a new generation of NA inhibitors via contemporary medicinal chemistry strategies to overcome drug resistance [18,19,20,21].

Except for N10 and N11 with no or extremely low sialidase activity, the other nine NAs are categorized into two phylogenetic groups: group 1 NA consists of N1, N4, N5 and N8 subtypes, whereas group 2 NA contains N2, N3, N6, N7 and N9 subtypes [22,23]. In group-1 NAs, with the exception of the 2009 pandemic H1N1 NA [24], a cavity known as the 150-cavity, consisting of residues 147–152, adopts a conformation adjacent to the active site [25]. In contrast, the 150-cavity of group-2 NAs is present in a closed conformation (Figure 2) [26]. The discovery of the 150-cavity in group-1 NAs provides a new strategy for the development of novel NA inhibitors with increased specificity and potency. In recent years, several *N*-substituted oseltamivir derivatives **JMC32**, **JMC20I** [27], **JMC21h**, **JMC21g**, **JMC17f** [28], **EJMC5c**, **EJMC13c [29]** and **JMC23d [30]**, **BMC17 [31]**, **EJMC8a [32]**, **JMC2 [33]** and **RSC2j** [34] (Figure 3) targeting the 150-cavity have been reported. Most of these inhibitors displayed better or similar inhibitory activities than **OSC** against both wild-type and mutant NAs. For instance, **JMC32**, **JMC20I**, **JMC21h**, **JMC21g** and **JMC23d** showed more potent activity than **OSC** against H5N1 with IC_50_ values of 2.1 nM, 1.9 nM, 0.96 nM, 3.83 nM and 0.27 nM, respectively. Moreover, most of the compounds showed in Figure 3 exhibited greater inhibitory activities than **OSC** against N5N1-H274Y. The discovery of these inhibitors proved that targeting 150-cavity was an effective strategy to enhance the efficacy of **OSC** derivatives against NAs.

In previous studies, our research efforts explored the substitution on the benzyl group by introducing different hydrophobic groups at the *para*-position of the benzyl group and led to the discovery of potent inhibitors **EJMC5c** and **EJMC13c**. The biological activity of **EJMC5c** and **EJMC13c** against H5N1-H274Y NA was 4.85- and 5.71-fold more potent than **OSC** with IC_50_ values of 0.33 μM and 0.28 μM, respectively. The present study is an extension of our previous work. Herein, we have kept benzoloxybenzyl unchanged and replaced borate ester with boronic acids, carboxylic acids, nitro groups, amides, sulfonamides and sulfonic acids by utilizing a bioisosteric replacement strategy to obtain novel oseltamivir derivatives, aiming to further explore structure–activity relationships in the 150-cavity (Figure 4). All newly synthesized compounds were evaluated in vitro for their neuraminidase inhibitory activities against H1N1, H3N2, H5N1, H5N8 and H5N1-H274Y. The cell-based antiviral activity and cytotoxicity of selected compounds were also characterized. Furthermore, based on the biological results, we performed molecular docking and predicted the effects of representative compounds on inhibition of CYP enzymes and safety assessment.

## 2. Results and Discussion

### 2.1. Chemistry

The C-5 *N*-substituted oseltamivir derivatives (**1c**–**18c**) were prepared via a well-established synthetic route as outlined in Figure 1. The target compounds **1c**–**18c** were synthesized from the commercially available starting material oseltamivir phosphate. Initially, 4-hydroxybenzaldehyde was treated with corresponding benzyl bromide in the presence of potassium carbonate to obtain the intermediates **1a**–**18a**. Then, treatment of oseltamivir phosphate with a series of different aldehydes in the presence of NaBH_3_CN yielded the intermediates **1b–18b**. The intermediates **1b**–**18b** were finally hydrolyzed with NaOH aqueous solution and acidified by HCl aqueous solution to obtain the target compounds **1c**–**18c**. All novel synthesized compounds were fully characterized by electrospray ionization mass spectrometry (ESI-MS), proton nuclear magnetic resonance spectroscopy (^1^H NMR) (Appendix A), as well as carbon nuclear magnetic resonance spectroscopy (^13^C NMR).

### 2.2. Biological Activity

#### 2.2.1. In Vitro Inhibitory Activities of Influenza Virus Neuraminidase

The synthesized novel oseltamivir derivatives were screened on NAs from H1N1, H3N2, H5N1, H5N8 and H5N1-H274Y according to our previous method [28,30,35]. The inhibition potencies of test compounds and reference compound oseltamivir carboxylate (OSC), are summarized in Table 1. Indeed, OSC showed a greater inhibitory potency toward wild-type NAs with IC_50_ values of 1.1 × 10^−3^ μM, 5.2 × 10^−3^ μM, 0.019 μM and 7.0 × 10^−3^ μM against H1N1, H3N2, H5N1 and H5N8, respectively. In line with the reported data, OSC exhibited weaker inhibitory activity against mutant H5N1-H274Y (IC_50_ = 1.25 μM) compared to that against wild-type H5N1 (IC_50_ = 0.019 μM).

As reported in Table 1, all the target compounds displayed decreased activities compared to that of **OSC** towards all wide-type NAs of group-1 (H1N1, H5N1 and H5N8) and group-2 (H3N2). With the exception of compounds **3c**, **7c** and **14c**, the other compounds showed better inhibitory activities against N1 (H1N1, H5N1) and N8 (H5N8) than N2 (H3N2). Intriguingly, compounds with R groups of 4-B(OH)_2_ (**3c**), 2-OMe, 5-COOH (**7c**), 3-CONH_2_ (**14c**) demonstrated greater or similar activities against N2 (H3N2, IC_50_ = 0.058 μM, 8.19 μM and 0.027 μM, respectively) relative to N1 (H1N1, IC_50_ = 0.045 μM, 14.02 μM and 0.011 μM, respectively), which might be explained by their ability to induce the opening of the 150-loop of group-2 NA.

Regrettably, compounds bearing *ortho*-boronic acid (**1c**), carboxylic acid (**4c**-**7c**, **9c-10c**), nitro (**11c**, **13c**), *para*-amide (**15c**), *meta*-sulfonamide (**16c**) and *para*-sulfonic acid (**18c**) showed sharply decreased potency against NAs compared to that of **OSC**. Interestingly, **8c** featuring 2-Cl, 4-COOH-substituted phenyl group (IC_50_ = 0.036 μM) displayed relatively similar activity to that of **OSC** (IC_50_ = 7.0 × 10^−3^ μM) against H5N8. Besides, **12c** (R = 3-NO_2_) with IC_50_ value of 0.023 μM exhibited equivalent inhibitory activity to that of **OSC** with IC_50_ value of 0.019 μM against H5N1. Additionally, compound **17c** carrying *para*-sulfonamide displayed comparable activity (IC_50_ = 0.027 μM, 0.053 μM, respectively) to **OSC** (IC_50_ = 1.1 × 10^−3^ μM, 0.019 μM, respectively) against H1N1 and H5N1. Notably, **3c** (R = 4-B(OH)_2_) and **14c** (R = 3-CONH_2_) exerted slightly weaker or equal inhibitory activities than **OSC** with IC_50_ values at the two-digit nanomolar range against both group-1 NAs (H1N1, H5N1 and H5N8) and group-2 NA (H3N2), while **2c** (R = 3-B(OH)_2_) proved to be potent inhibitor of group-1 NAs (H1N1, H5N1 and H5N8).

The inhibitory activity of all compounds against mutant H5N1-H274Y NA were also tested. As shown in Table 1, it was noticed that most of the compounds (except for **2c**, **3c**, **12c**, **14c** and **17c**) were found to be less active against the mutant H5N1-H274Y NA compared to that of **OSC**. The inhibitory activities of compounds **12c** and **17c** (with IC_50_ values of 0.87 μM and 0.85 μM, respectively) were close to that of **OSC** (with IC_50_ value of 1.25 μM). Impressively, **2c**, **3c** and **14c** (IC_50_ = 0.27 μM, 0.56 μM and 0.57 μM, respectively) were 2.2- to 4.6-fold more potent than **OSC**. Indeed, these compounds could be used as lead compounds for further optimization.

#### 2.2.2. In Vitro Anti-Influenza Virus Activity

Considering the enzymatic inhibition activity results, the promising compounds **2c**, **3c** and **14c** were further evaluated for antiviral activity and cytotoxicity in Chicken Embryo Fibroblast cells (CEFs) infected with A/Goose/Guangdong/SH7/2013 (H5N1) as well as A/Goose/Jiangsu/1306/2014 (H5N8). Oseltamivir carboxylate (**OSC**) was used as a reference compound in parallel. The values of EC_50_ (anti-influenza virus activity) and CC_50_ (cytotoxicity) of the selected compounds were determined.

As outlined in Table 2, all the tested compounds showed no appreciable cytotoxicity in CEFs. In the case of H5N1 and H5N8 viruses, compounds **3c** (EC_50_ = 3.20 μM and 2.51 μM, respectively) and **14c** (EC_50_ = 1.18 μM and 2.80 μM, respectively) displayed moderate activities compared to that of **OSC** (EC_50_ = 0.76 μM and 2.01 μM, respectively). Notably, compound **2c** (EC_50_ = 0.69 μM and 1.57 μM, respectively) exhibited the most potent inhibitory activity, which was better than **OSC** against H5N1 and H5N8.

Then, the representative compounds **2c**, **3c** and **14c** were also tested against A/PR/8/34 (H1N1) and A/Wisconsin/67/05 (H3N2) strains in MDCK cells by means of a plaque reduction assay (PRA). The cytotoxicity was carried out in MDCK cells using the 3-(4,5-dimethylthiazol-2-yl)-2,5-diphenyltetrazolium bromide (MTT) method. **OSC** and zanamivir (**ZAN**) were included as controls. The EC_50_ and CC_50_ values are illustrated in Table 3.

For the test compounds, no cytotoxicity was observed up to the highest tested concentration (250 μM) in MDCK cells. As for the H1N1 strain, compounds **2c** and **3c** exhibited nearly equivalent potency with EC_50_ values of 0.03 μM and 0.02 μM, respectively, compared to that of **OSC** (EC_50_ = 0.02 μM) and **ZAN** (EC_50_ = 9 × 10^−3^ μM). In contrast, **14c** (EC_50_ = 0.21 μM) exerted decreased activity. Against H3N2 virus, all the tested compounds exhibited weaker potency (EC_50_ values ranging from 3.08 μM to > 100 μM) compared to that of **OSC** and **ZAN**. The antiviral activities of compounds **3c** against H3N2 and **14c** against H1N1 and H3N2 were inferior to their NA-inhibitory potency, which could be due to the cellular metabolism of the molecules and/or poor membrane permeability.

### 2.3. Molecular Docking

In order to further understand the binding modes of novel oseltamivir derivatives in the 150-cavity of group-1 NAs, molecular docking studies of representative compound **2c** to crystal structures of N1 (PDB ID: 2HU0) and N1-H274Y (PDB ID: 3CL0) were performed by using Schrödinger Maestro 12.9 software and the docking results were visualized by PyMOL.

From Figure 5, it can be observed that the 4-(3-boronic acid benzyloxy)benzyl group of compound **2c** could extend into the 150-cavity of N1 (H5N1, Figure 5A) and N1-H274Y (H5N1-H274Y, Figure 5D), while the oseltamivir carboxylic acid part of this compound was well embedded in the active sites of NAs in line with the binding pattern of **OSC**. In Figure 5B,C, it could be found that **2c** retained key hydrogen bonds between carboxylic acid and Arg292, Arg371 and Arg118, which was consistent with **OSC** binding. Moreover, an additional H-bond interaction was formed between the terminal boronic acid of **2c** and NH_2_ of Arg118. However, the hydrogen bonds formed by C-1 carboxyl acid and C-4 acetamide of **OSC** with Tyr347 and Arg152, respectively, disappeared in **2c**, which might be the reason why **2c** showed 3.9-fold lower potency against H5N1 in the enzymatic assay. 

According to previous research, the bulker tyrosine residue pushes the carboxyl group of Glu276 close to the binding site due to the mutation of His274Tyr. In this position, the charged carboxyl group destroys the otherwise hydrophobic pocket that normally accommodates the pentyloxy substituent of oseltamivir, thus causing a change in the conformation of **OSC**, which explained the resistance of the N1-H274Y mutant to oseltamivir. From Figure 5F, it was found that the hydrogen bonds formed by the C-4 amide groups of **OSC** with Arg152 disappeared, possibly due to the change in the conformation of **OSC**. However, as shown in Figure 5E, it could be found that **2c** retained key hydrogen bonds between amide groups and Arg152, which may be due to the introduction of 4-(3-boronic acid benzyloxy)benzyl group, which extends into the 150-cavity and further distorts the conformation of the parent ring. Although C-1 carboxylic acid and C-5 amino of **OSC** generated additional hydrogen bonds with Tyr263, Tyr321 and Glu38, respectively, new hydrogen bonds compensated for the lost interactions built by boronic acid of **2c** with Gln136 and Val149, and by C-4 acetamide of **2c** with Arg152. Meanwhile, benzyl of **2c** might produce hydrophobic interactions with Thr439, Val116 and Asp151. This might account for the superior inhibition of **2c** against H5N1-H274Y NA compared to that of **OSC**. Overall, it should be noted that the molecule docking results were consistent with our original design hypothesis that the modification targeting of the 150-cavity of NAs is an effective strategy for overcoming drug resistance profile.

### 2.4. In Silico Predicted Effects of Representative Compounds on Inhibition of CYP Enzymes

The important metabolic enzyme, cytochrome P450 (CYP), exerts a significant influence in drug metabolism. However, several drugs can also induce or inhibit CYP enzymes, and therefore lead to a decreased therapeutic effect or an adverse reaction profile when co-administrating different drugs, thus causing metabolic-mediated drug–drug interactions (DDI). Thus, we evaluated the inhibitory potential of compounds **2c**, **3c**, **14c**, **EJMC5c**, **EJMC13c** and **OSC** on the main CYP enzymes by utilizing online software (http://admet.scbdd.com/calcpre/index/). As summarized in Table 4, compounds **3c** and **14c** were not predicted to inhibit CYP1A2, CYP2C9, CYP2C19, CYP2D6 and CYP3A4. In addition, **2c** displayed similar CYP enzymatic inhibition of the five CYP isozymes compared to that of **EJMC5c**.

### 2.5. In Silico Prediction of Physicochemical Properties

Additionally, the drug-like properties of representative compounds **2c**, **3c**, **14c**, **EJMC5c**, **EJMC13c** and **OSC** were characterized using a free online software (http://www.molinspiration.com/, accessed on 6 September 2022). As shown in Table 5, the results indicated that various parameters of compounds **2c**, **3c** and **14c** including hydrogen bond acceptors (nON), topological polar surface area (tPSA) and Molinspiration-predicted Log P (miLog P) were all in the acceptable range, except for the slight deviation of hydrogen bond donors (nOHNH) and rotatable bonds (nrotb). Inspiringly, all of the test compounds displayed an acceptable miLog P, while the **OSC** went beyond the normal criteria. The tPSA values of all compounds were in the range of 101.65 to 139.99 Å^2^, suggesting their advantage for intestinal absorption (<140 Å^2^) and inability to pass through the blood–brain barrier (>60 Å^2^).

### 2.6. Safety Assessment

A single-dose toxicity assay of compound **2c** was conducted in healthy Kunming mice. After intragastric administration of **2c** at a dose of 1000 mg·kg^−1^, no death occurred, and no significant behavior abnormalities (lethargy, clonic convulsions, anorexia, and ruffled fur) were observed. As illustrated in Figure 6, the body weights of both female and male mice increased gradually within a week. Thus, this would argue that **2c** was well-tolerated at a dose of 1000 mg/kg with no acute toxicity.

## 3. Conclusions

In summary, we designed and synthesized a novel series of *N*-substituted oseltamivir derivatives using a bioisosterism strategy for further structural exploration of the 150-cavity. Among the tested series, compounds **2c**, **3c** and **14c** exhibited moderate to remarkable inhibitory potency against N1 (H1N1), N1 (H5N1), N8 (H5N8) and mutant N1 (H5N1-H274Y) compared to that of **OSC**. In particular, **2c** (IC_50_ = 0.27 μM) showed 4.6-fold more potent activity than **OSC** (IC_50_ = 1.25 μM) toward H5N1-H274Y NA. Furthermore, **2c** maintained similar activity against H1N1 in MDCK cells, which reached the same level to that of **OSC**, and also exhibited a slightly improved potency against H5N1 and H5N8 in CEFs over that of **OSC**. Meanwhile, **2c** displayed low cytotoxicity in CEFs and MDCK cells and no acute toxicity in Kunming mice. Molecular docking studies were carried out to rationalize our design hypotheses and offer guidance for further structural optimizations. Collectively, we consider **2c** is a valuable lead compound and further modification is in progress.

## 4. Experimental Section

### 4.1. Chemistry

The key chemical reactant oseltamivir phosphate and the other chemicals and reagents were purchased from commercial suppliers and used without further purification. Solvents were of reagent grade and were purified and dried with standard methods when necessary. Thin-layer chromatography (TLC) was performed on Silica Gel GF254 for TLC, and spots were visualized by irradiation with UV light (λ = 254 nm and 365 nm). Flash column chromatography was carried out on columns packed with Silica Gel 60 (200–300 mesh), purchased from Qingdao Haiyang Chemical Company. A rotary evaporator was used for the concentration of the reaction solutions under reduced pressure. All melting points (mp) were determined on a micro melting-point apparatus (RY-1G; Tianjin TianGuang Optical Instruments) and were uncorrected. The ^1^H NMR and ^13^C NMR spectra were recorded on a Bruker AV-400 spectrometer in the indicated solvent DMSO-*d*_6_ and CD_3_OD with tetramethylsilane (TMS) as the internal standard. Chemical shifts were expressed in *δ* units (ppm) and *J* values were presented in hertz (Hz). Mass spectra (MS) were obtained on an API 4000 LC/MS spectrometer (Applied Biosystems, Foster City, CA, USA).

#### 4.1.1. General Procedure for the Preparation of Compounds **1a**–**18a**

A solution of 4-hydroxybenzaldehyde (1 g, 8.2 mmol), substituted benzyl bromide (8.4 mmol, 1.02 eq) and potassium carbonate (1.7 g, 12.3 mmol) in 30 mL of *N,N*-dimethylformamide (DMF) was stirred overnight under an inert atmosphere at room temperature [29]. Subsequently, 50 mL water was added and the mixture was extracted with ethyl acetate (40 mL × 3). The combined organic phase was washed with a saturated brine (30 mL × 2), dried over anhydrous MgSO_4_, filtered, and concentrated under reduced pressure to give the crude product, which was purified by column chromatography on silica gel (10–50% ethyl acetate in petroleum ether) to obtain the corresponding products, **1a**–**18a**. 

(2-((4-Formylphenoxy)methyl)phenyl)boronic acid (**1a**). White powder, 31%, mp: 185.1–186.7 °C. ^1^H NMR (400 MHz, DMSO-*d_6_*) δ: 9.83 (s, *J* = 27.6 Hz, 1H, CHO), 8.10 (s, 1H, OH), 7.93–7.51 (m, 3H, 3Ph-H), 7.51–6.98 (m, 5H, 5Ph-H), 5.58 (d, *J* = 15.7 Hz, 1H, CH), 5.36 (s, 1H, CH), 3.42 (s, 1H, OH).

(3-((4-Formylphenoxy)methyl)phenyl)boronic acid (**2a**). White powder, 24%, mp: 170.1–172.2 °C. ^1^H NMR (400 MHz, DMSO-*d_6_*) δ: 9.96–9.76 (m, *J* = 4.2 Hz, 1H, CHO), 8.09 (s, 1H, OH), 7.98–7.63 (m, 4H, 4Ph-H), 7.50 (d, *J* = 7.3 Hz, 1H, Ph-H), 7.39 (dt, *J* = 21.4, 6.6 Hz, 1H, Ph-H), 7.22 (dd, *J* = 6.6, 1.8 Hz, 2H, 2Ph-H), 5.22 (t, *J* = 24.0 Hz, 2H, CH_2_), 3.52 (s, 1H, OH).

(4-((4-Formylphenoxy)methyl)phenyl)boronic acid (**3a**). White powder, 38%, mp: 129.5–131.7 °C. ^1^H NMR (400 MHz, DMSO-*d*_6_) δ: 9.87 (s, 1H, CHO), 7.87 (d, *J* = 8.1 Hz, 2H, 2Ph-H), 7.81 (d, *J* = 7.5 Hz, 2H, 2Ph-H), 7.42 (d, *J* = 7.5 Hz, 2H, 2Ph-H), 7.21 (d, *J* = 8.2 Hz, 2H, 2Ph-H), 5.24 (s, 2H, CH_2_), 2.89 (s, 1H, OH), 2.73 (s, 1H, OH).

Methyl 2-((4-formylphenoxy)methyl)benzoate (**4a**). White powder, 62%, mp: 89.8–91.8 °C. ^1^H NMR (400 MHz, DMSO-*d_6_*) δ: 9.88 (s, 1H, CHO), 7.94 (d, *J* = 7.7 Hz, 1H, Ph-H), 7.89 (d, *J* = 8.7 Hz, 2H, 2Ph-H), 7.69–7.61 (m, 2H, 2Ph-H), 7.54–7.47 (m, 1H, Ph-H), 7.19 (d, *J* = 8.7 Hz, 2H, 2Ph-H), 5.54 (s, 2H, CH_2_), 3.81 (s, 3H, CH_3_).

Methyl 3-((4-formylphenoxy)methyl)benzoate (**5a**). White powder, 64%, mp: 79.1–81.2 °C. ^1^H NMR (400 MHz, DMSO-*d_6_*) δ: 9.88 (s, 1H, CHO), 8.07 (s, 1H, Ph-H), 7.95 (d, *J* = 7.8 Hz, 1H, Ph-H), 7.89 (d, *J* = 8.7 Hz, 2H, 2Ph-H), 7.76 (d, *J* = 7.7 Hz, 1H, Ph-H), 7.58 (t, *J* = 7.7 Hz, 1H, Ph-H), 7.23 (d, *J* = 8.6 Hz, 2H, 2Ph-H), 5.33 (s, 2H, CH_2_), 3.87 (s, 3H, CH_3_).

Methyl 4-((4-formylphenoxy)methyl)benzoate (**6a**). White powder, 61%, mp: 107.1–109.5 °C. ^1^H NMR (400 MHz, DMSO-*d_6_*) δ: 9.88 (s, 1H, CHO), 8.00 (d, *J* = 8.2 Hz, 2H, 2Ph-H), 7.89 (d, *J* = 8.7 Hz, 2H, 2Ph-H), 7.62 (d, *J* = 8.1 Hz, 2H, 2Ph-H), 7.23 (d, *J* = 8.6 Hz, 2H, 2Ph-H), 5.35 (s, 2H, CH_2_), 3.86 (s, 3H, CH_3_).

Methyl 3-((4-formylphenoxy)methyl)-4-methoxybenzoate (**7a**). White powder, 76%, mp: 66.5–68.1 °C. ^1^H NMR (400 MHz, DMSO-*d_6_*) δ: 9.89 (s, 1H, CHO), 8.06–7.97 (m, 2H, 2Ph-H), 7.89 (d, *J* = 8.6 Hz, 2H, 2Ph-H), 7.22 (d, *J* = 8.5 Hz, 3H, 3Ph-H), 5.23 (s, 2H, CH_2_), 3.93 (s, 3H, CH_3_), 3.82 (s, 3H, CH_3_).

Methyl 3-chloro-4-((4-formylphenoxy)methyl)benzoate (**8a**). White powder, 73%, mp: 109.8–111.5 °C. ^1^H NMR (400 MHz, DMSO-*d_6_*) δ: 9.90 (s, 1H, CHO), 8.01 (d, *J* = 1.2 Hz, 1H, Ph-H), 7.97 (dd, *J* = 8.0, 1.3 Hz, 1H, Ph-H), 7.91 (d, *J* = 8.7 Hz, 2H, 2Ph-H), 7.78 (d, *J* = 8.0 Hz, 1H, Ph-H), 7.26 (d, *J* = 8.7 Hz, 2H, 2Ph-H), 5.37 (s, 2H, CH_2_), 3.89 (s, 3H, CH_3_).

Methyl 4-((4-formylphenoxy)methyl)-3-methoxybenzoate (**9a**). White powder, 72%, mp: 92.2–93.8 °C. ^1^H NMR (400 MHz, DMSO-*d_6_*) δ: 9.88 (s, 1H, CHO), 7.89 (d, *J* = 8.7 Hz, 2H, 2Ph-H), 7.64–7.53 (m, *J* = 19.5, 6.1 Hz, 3H, 3Ph-H), 7.21 (d, *J* = 8.6 Hz, 2H, 2Ph-H), 5.26 (s, 2H, CH_2_), 3.92 (s, 3H, CH_3_), 3.87 (s, 3H, CH_3_).

Methyl 4-((4-formylphenoxy)methyl)-2-methoxybenzoate (**10a**). White powder, 72%, mp: 79.1–81.5 °C. ^1^H NMR (400 MHz, DMSO-*d_6_*) δ: 9.88 (s, 1H, CHO), 7.89 (d, *J* = 8.7 Hz, 2H, 2Ph-H), 7.68 (d, *J* = 7.9 Hz, 1H, Ph-H), 7.27 (s, 1H, Ph-H), 7.23 (d, *J* = 8.6 Hz, 2H, 2Ph-H), 7.11 (d, *J* = 7.9 Hz, 1H, Ph-H), 5.29 (s, 2H, CH_2_), 3.84 (s, 3H, CH_3_), 3.79 (s, 3H, CH_3_).

4-((2-Nitrobenzyl)oxy)benzaldehyde (**11a**). White powder, 62%, mp: 85.1–87.2 °C. ^1^H NMR (400 MHz, DMSO-*d_6_*) δ: 9.88 (s, 1H, CHO), 8.15 (d, *J* = 8.1 Hz, 1H, Ph-H), 7.92–7.87 (m, 2H, 2Ph-H), 7.84–7.77 (m, 2H, 2Ph-H), 7.65 (ddd, *J* = 8.6, 6.8, 4.1 Hz, 1H, Ph-H), 7.22 (d, *J* = 8.7 Hz, 2H, 2Ph-H), 5.59 (s, 2H, CH_2_).

4-((3-Nitrobenzyl)oxy)benzaldehyde (**12a**). White powder, 66%, mp: 79.7–81.5 °C. ^1^H NMR (400 MHz, DMSO-*d_6_*) δ: 9.89 (s, 1H, CHO), 8.35 (s, 1H, Ph-H), 8.22 (dd, *J* = 8.2, 1.4 Hz, 1H, Ph-H), 7.92 (dd, *J* = 17.7, 8.2 Hz, 3H, 3Ph-H), 7.73 (t, *J* = 7.9 Hz, 1H, Ph-H), 7.26 (d, *J* = 8.7 Hz, 2H, 2Ph-H), 5.41 (s, 2H, CH_2_).

4-((4-Nitrobenzyl)oxy)benzaldehyde (**13a**). Yellow powder, 65%. ^1^H NMR (400 MHz, DMSO-*d_6_*) δ: 9.89 (s, 1H, CHO), 8.28 (d, *J* = 8.7 Hz, 2H, 2Ph-H), 7.90 (d, *J* = 8.7 Hz, 2H, 2Ph-H), 7.75 (d, *J* = 8.6 Hz, 2H, 2Ph-H), 7.24 (d, *J* = 8.7 Hz, 2H, 2Ph-H), 5.42 (s, 2H, CH_2_).

3-((4-Formylphenoxy)methyl)benzamide (**14a**). Pale yellow oil, 63% yield. ^1^H NMR (400 MHz, DMSO-*d_6_*) δ: 9.87 (s, 1H, CHO), 8.00 (d, *J* = 12.9 Hz, 2H, 2Ph-H), 7.87 (dd, *J* = 11.6, 8.6 Hz, 3H, Ph-H, NH_2_), 7.62 (d, *J* = 7.6 Hz, 1H, Ph-H), 7.49 (t, *J* = 7.7 Hz, 1H, Ph-H), 7.41 (s, 1H, Ph-H), 7.23 (d, *J* = 8.6 Hz, 2H, 2Ph-H), 5.28 (s, 2H, CH_2_).

4-((4-Formylphenoxy)methyl)benzamide (**15a**). White powder, 73%, mp: 185.8–188.2 °C. ^1^H NMR (400 MHz, DMSO-*d_6_*) δ: 9.88 (s, 1H, CHO), 7.99 (s, 1H, NH), 7.89 (t, *J* = 7.8 Hz, 4H, 4Ph-H), 7.54 (d, *J* = 8.1 Hz, 2H, 2Ph-H), 7.39 (s, 1H, NH), 7.22 (d, *J* = 8.7 Hz, 2H, 2Ph-H), 5.31 (s, 2H, CH_2_).

3-((4-Formylphenoxy)methyl)benzenesulfonamide (**16a**). White powder, 67%, mp: 137.2–139.7 °C. ^1^H NMR (400 MHz, DMSO-*d_6_*) δ: 9.89 (s, 1H, CHO), 7.95 (s, 1H, Ph-H), 7.90 (d, *J* = 8.8 Hz, 2H, 2Ph-H), 7.82 (d, *J* = 7.8 Hz, 1H, Ph-H), 7.70 (d, *J* = 7.7 Hz, 1H, Ph-H), 7.62 (t, *J* = 7.7 Hz, 1H, Ph-H), 7.42 (s, 2H, NH_2_), 7.24 (d, *J* = 8.7 Hz, 2H, 2Ph-H), 5.34 (s, 2H, CH_2_).

4-((4-Formylphenoxy)methyl)benzenesulfonamide (**17a**). White powder, 65%, mp: 124.5–126.5 °C. ^1^H NMR (400 MHz, DMSO-*d_6_*) δ: 9.87 (s, 1H, CHO), 7.87 (dd, *J* = 12.4, 8.5 Hz, 4H, 4Ph-H), 7.65 (d, *J* = 8.2 Hz, 2H, 2Ph-H), 7.37 (s, 2H, NH_2_), 7.22 (d, *J* = 8.6 Hz, 2H, 2Ph-H), 5.34 (s, 2H, CH_2_).

4-((4-Formylphenoxy)methyl)benzenesulfonyl fluoride (**18a**). White powder, 72%, mp: 74.8–76.2 °C. ^1^H NMR (400 MHz, DMSO-*d_6_*) δ: 9.89 (s, 1H, CHO), 8.19 (d, *J* = 8.4 Hz, 2H, 2Ph-H), 7.89 (dd, *J* = 16.0, 8.4 Hz, 4H, 4Ph-H), 7.25 (d, *J* = 8.6 Hz, 2H, 2Ph-H), 5.47 (s, 2H, CH_2_).

#### 4.1.2. General Procedure for the Preparation of Compounds **1b**–**18b**

To a solution of oseltamivir phosphate (0.82 g, 2.0 mmol) in 30 mL methanol, different substituted aldehyde (2.0 mmol, 1 eq) was added at room temperature. The reaction mixture was stirred for 0.5 h, and then NaBH_3_CN (0.31 g, 5.0 mmol, 2.5 eq) was added. The resulting mixture was stirred at room temperature until completion, monitored by TLC. After removal of the excess solvent under reduced pressure, saturated brine (30 mL) and saturated sodium carbonate solution (10 mL) were added. The aqueous phase was extracted with ethyl acetate (3 × 30 mL). Then, the combined organic phase was dried over anhydrous MgSO_4_, filtered, and purified by flash column chromatography (0–6% methanol in dichloromethane) to provide the corresponding intermediate **1b–18b**.

(2-((4-((((1S,5R,6R)-6-acetamido-3-(ethoxycarbonyl)-5-(pentan-3-yloxy)cyclohex-3-en-1-yl)amino)methyl)phenoxy)methyl)phenyl)boronic acid (**1b**). Pale yellow powder, 74% yield, mp: 132.5–134.7 °C. ^1^H NMR (400 MHz, CD_3_OD) δ: 7.44–7.21 (m, 6H, 6Ph-H), 6.99 (d, *J* = 7.9 Hz, 2H, 2Ph-H), 6.78 (s, 1H, CH), 5.10 (s, 2H, CH_2_), 4.21 (q, *J* = 7.0 Hz, 2H, CH_2_), 4.07 (d, *J* = 10.3 Hz, 1H, CH), 3.93 (t, *J* = 9.4 Hz, 1H, CH), 3.86 (d, *J* = 12.7 Hz, 1H, CH), 3.70 (d, *J* = 12.6 Hz, 1H, CH), 3.42–3.36 (m, 1H, CH), 2.95 (td, *J* = 10.0, 5.4 Hz, 1H, CH), 2.83 (dd, *J* = 17.7, 5.0 Hz, 1H, CH), 2.34–2.20 (m, 1H, CH), 2.00 (s, 3H, CH_3_), 1.59–1.42 (m, 4H, 2CH_2_), 1.29 (t, *J* = 7.0 Hz, 3H, CH_3_), 0.90 (q, *J* = 8.0 Hz, 6H, 2CH_3_). ESI-MS: *m/z* 553.3 [M + H]^+^, C_30_H_41_BN_2_O_7_ (552.48).

(3-((4-((((1S,5R,6R)-6-acetamido-3-(ethoxycarbonyl)-5-(pentan-3-yloxy)cyclohex-3-en-1-yl)amino)methyl)phenoxy)methyl)phenyl)boronic acid (**2b**). White powder, 73% yield, mp: 163.1–165.2 °C. ^1^H NMR (400 MHz, CD_3_OD) δ: 7.84–7.70 (m, 1H, Ph-H), 7.70–7.57 (m, 1H, Ph-H), 7.45 (d, *J* = 7.4 Hz, 1H, Ph-H), 7.33 (t, *J* = 7.5 Hz, 1H, Ph-H), 7.25 (d, *J* = 7.8 Hz, 2H, 2Ph-H), 6.97 (d, *J* = 7.7 Hz, 2H, 2Ph-H), 6.78 (s, 1H, CH), 5.07 (s, 2H, CH_2_), 4.22 (q, *J* = 6.9 Hz, 2H, CH_2_), 4.07 (d, *J* = 8.5 Hz, 1H, CH), 3.94 (t, *J* = 9.5 Hz, 1H, CH), 3.88 (d, *J* = 12.6 Hz, 1H, CH), 3.71 (d, *J* = 12.7 Hz, 1H, CH), 3.39 (q, *J* = 5.1 Hz, 1H, CH), 2.99 (dt, *J* = 15.1, 7.8 Hz, 1H, CH), 2.84 (dd, *J* = 17.5, 5.0 Hz, 1H, CH), 2.35–2.23 (m, 1H, CH), 1.99 (s, 3H, CH_3_), 1.57–1.44 (m, 4H, 2CH_2_), 1.29 (t, *J* = 6.9 Hz, 3H, CH_3_), 0.89 (q, *J* = 8.0 Hz, 6H, 2CH_3_). ESI-MS: *m/z* 553.5 [M + H]^+^, C_30_H_41_BN_2_O_7_ (552.48).

(4-((4-((((1S,5R,6R)-6-acetamido-3-(ethoxycarbonyl)-5-(pentan-3-yloxy)cyclohex-3-en-1-yl)amino)methyl)phenoxy)methyl)phenyl)boronic acid (**3b**). White powder, 69% yield, mp: 189.2–191.1 °C. ^1^H NMR (400 MHz, CD_3_OD) δ: 7.70 (s, 2H, 2Ph-H), 7.39 (d, *J* = 7.5 Hz, 2H, 2Ph-H), 7.25 (d, *J* = 7.9 Hz, 2H, 2Ph-H), 6.96 (d, *J* = 7.9 Hz, 2H, 2Ph-H), 6.79 (s, 1H, CH), 5.08 (s, 2H, CH_2_), 4.22 (q, *J* = 7.0 Hz, 2H, CH_2_), 4.08 (d, *J* = 7.9 Hz, 1H, CH), 4.00–3.85 (m, 2H, 2CH), 3.73 (d, *J* = 12.7 Hz, 1H, CH), 3.42–3.36 (m, 1H, CH), 3.00 (dt, *J* = 15.1, 7.7 Hz, 1H, CH), 2.84 (dd, *J* = 17.4, 5.0 Hz, 1H, CH), 2.30 (dd, *J* = 17.4, 9.9 Hz, 1H, CH), 2.00 (s, 3H, CH_3_), 1.59–1.42 (m, 4H, 2CH_2_), 1.29 (t, *J* = 7.0 Hz, 3H, CH_3_), 0.89 (q, *J* = 8.0 Hz, 6H, 2CH_3_). ESI-MS: *m/z* 553.4 [M + H]^+^, C_30_H_41_BN_2_O_7_ (552.48).

Methyl 2-((4-((((1S,5R,6R)-6-acetamido-3-(ethoxycarbonyl)-5-(pentan-3-yloxy)cyclohex-3-en-1-yl)amino)methyl)phenoxy)methyl)benzoate (**4b**). White powder, 68% yield, mp: 72.7–74.1 °C. ^1^H NMR (400 MHz, CD_3_OD) δ: 7.97 (d, *J* = 7.8 Hz, 1H, Ph-H), 7.69 (d, *J* = 7.8 Hz, 1H, Ph-H), 7.56 (t, *J* = 7.6 Hz, 1H, Ph-H), 7.41 (t, *J* = 7.6 Hz, 1H, Ph-H), 7.24 (d, *J* = 8.5 Hz, 2H, 2Ph-H), 6.93 (d, *J* = 8.6 Hz, 2H, 2Ph-H), 6.77 (s, 1H, CH), 5.43 (s, 2H, CH_2_), 4.21 (q, *J* = 7.1 Hz, 2H, CH_2_), 4.05 (d, *J* = 8.5 Hz, 1H, CH), 3.94–3.85 (m, 4H, CH, CH_3_), 3.80 (d, *J* = 12.7 Hz, 1H, CH), 3.64 (d, *J* = 12.7 Hz, 1H, CH), 3.37 (dt, *J* = 11.2, 5.6 Hz, 1H, CH), 2.94–2.85 (m, 1H, CH), 2.81 (dd, *J* = 17.6, 5.0 Hz, 1H, CH), 2.28–2.18 (m, 1H, CH), 1.99 (s, 3H, CH_3_), 1.56–1.45 (m, 4H, 2CH_2_), 1.29 (t, *J* = 7.1 Hz, 3H, CH_3_), 0.89 (dt, *J* = 9.9, 7.4 Hz, 6H, 2CH_3_). ESI-MS: *m/z* 567.6 [M + H]^+^, C_32_H_42_N_2_O_7_ (566.70).

Methyl 3-((4-((((1S,5R,6R)-6-acetamido-3-(ethoxycarbonyl)-5-(pentan-3-yloxy)cyclohex-3-en-1-yl)amino)methyl)phenoxy)methyl)benzoate (**5b**). White powder, 72% yield, mp: 75.2–77.1 °C. ^1^H NMR (400 MHz, CD_3_OD) δ: 8.09 (s, 1H, Ph-H), 7.96 (d, *J* = 7.8 Hz, 1H, Ph-H), 7.68 (d, *J* = 7.6 Hz, 1H, Ph-H), 7.49 (t, *J* = 7.7 Hz, 1H, Ph-H), 7.25 (d, *J* = 8.5 Hz, 2H, 2Ph-H), 6.97 (d, *J* = 8.5 Hz, 2H, 2Ph-H), 6.78 (s, 1H, CH), 5.13 (s, 2H, CH_2_), 4.21 (q, *J* = 7.1 Hz, 2H, CH_2_), 4.06 (d, *J* = 8.2 Hz, 1H, CH), 3.97–3.87 (m, 4H, CH, CH_3_), 3.84 (d, *J* = 12.7 Hz, 1H, CH), 3.67 (d, *J* = 12.7 Hz, 1H, CH), 3.41–3.34 (m, 1H, CH), 2.93 (d, *J* = 5.5 Hz, 1H, CH), 2.83 (d, *J* = 17.6 Hz, 1H, CH), 2.31–2.20 (m, 1H, CH), 1.99 (s, 3H, CH_3_), 1.57–1.44 (m, 4H, 2CH_2_), 1.29 (t, *J* = 7.1 Hz, 3H, CH_3_), 0.89 (dd, *J* = 17.0, 7.5 Hz, 6H, 2CH_3_). ESI-MS: *m/z* 567.10 [M + H]^+^, C_32_H_42_N_2_O_7_ (566.70).

Methyl 4-((4-((((1S,5R,6R)-6-acetamido-3-(ethoxycarbonyl)-5-(pentan-3-yloxy)cyclohex-3-en-1-yl)amino)methyl)phenoxy)methyl)benzoate (**6b**). White powder, 67% yield, mp: 141.5–143.5 °C. ^1^H NMR (400 MHz, CD_3_OD) δ: 8.01 (d, *J* = 8.2 Hz, 2H, 2Ph-H), 7.54 (d, *J* = 8.2 Hz, 2H, 2Ph-H), 7.24 (d, *J* = 8.5 Hz, 2H, 2Ph-H), 6.96 (d, *J* = 8.6 Hz, 2H, 2Ph-H), 6.77 (s, 1H, CH), 5.15 (s, 2H, CH_2_), 4.21 (q, *J* = 7.1 Hz, 2H, CH_2_), 4.06 (d, *J* = 7.9 Hz, 1H, CH), 3.95–3.87 (m, 4H, CH, CH_3_), 3.81 (d, *J* = 12.7 Hz, 1H, CH), 3.65 (d, *J* = 12.7 Hz, 1H, CH), 3.41–3.34 (m, 1H, CH), 2.90 (td, *J* = 10.0, 5.4 Hz, 1H, CH), 2.81 (dd, *J* = 17.6, 5.1 Hz, 1H, CH), 2.24 (ddt, *J* = 15.1, 9.4, 2.6 Hz, 1H, CH), 1.99 (s, 3H, CH_3_), 1.55–1.45 (m, 4H, 2CH_2_), 1.29 (t, *J* = 7.1 Hz, 3H, CH_3_), 0.89 (dt, *J* = 9.8, 7.4 Hz, 6H, 2CH_3_). ESI-MS: *m/z* 567.5 [M + H]^+^, C_32_H_42_N_2_O_7_ (566.70).

Methyl 3-((4-((((1S,5R,6R)-6-acetamido-3-(ethoxycarbonyl)-5-(pentan-3-yloxy)cyclohex-3-en-1-yl)amino)methyl)phenoxy)methyl)-4-methoxybenzoate (**7b**). Colorless sticky oil, 70% yield. ^1^H NMR (400 MHz, CD_3_OD) δ: 8.06 (d, *J* = 1.2 Hz, 1H, Ph-H), 8.00 (dd, *J* = 8.7, 1.8 Hz, 1H, Ph-H), 7.29 (d, *J* = 8.5 Hz, 2H, 2Ph-H), 7.10 (d, *J* = 8.7 Hz, 1H, Ph-H), 6.99 (d, *J* = 8.5 Hz, 2H, 2Ph-H), 6.80 (s, 1H, CH), 5.11 (s, 2H, CH_2_), 4.22 (q, *J* = 7.1 Hz, 2H, CH_2_), 4.10 (d, *J* = 8.1 Hz, 1H, CH), 4.05–3.93 (m, 5H, CH_2_, CH_3_), 3.86 (s, 3H, CH_3_), 3.82 (d, *J* = 12.9 Hz, 1H, CH), 3.43–3.36 (m, 1H, CH), 3.09 (td, *J* = 10.1, 5.6 Hz, 1H, CH), 2.88 (dd, *J* = 17.0, 4.7 Hz, 1H, CH), 2.42–2.30 (m, 1H, CH), 2.01 (s, 3H, CH_3_), 1.56–1.45 (m, 4H, 2CH_2_), 1.30 (t, *J* = 7.1 Hz, 3H, CH_3_), 0.89 (dd, *J* = 16.5, 7.5 Hz, 6H, 2CH_3_). ESI-MS: *m/z* 598.05 [M + H]^+^, C_33_H_44_N_2_O_8_ (596.72).

Methyl 4-((4-((((1S,5R,6R)-6-acetamido-3-(ethoxycarbonyl)-5-(pentan-3-yloxy)cyclohex-3-en-1-yl)amino)methyl)phenoxy)methyl)-3-chlorobenzoate (**8b**). White powder, 73% yield, mp: 159.5–161.5 °C. ^1^H NMR (400 MHz, CD_3_OD) δ: 8.03 (d, *J* = 1.1 Hz, 1H, Ph-H), 7.95 (dd, *J* = 8.1, 1.1 Hz, 1H, Ph-H), 7.69 (d, *J* = 8.0 Hz, 1H, Ph-H), 7.26 (d, *J* = 8.5 Hz, 2H, 2Ph-H), 6.97 (d, *J* = 8.5 Hz, 2H, 2Ph-H), 6.77 (s, 1H, CH), 5.21 (s, 2H, CH_2_), 4.21 (q, *J* = 7.1 Hz, 2H, CH_2_), 4.05 (d, *J* = 8.3 Hz, 1H, CH), 3.96–3.86 (m, 4H, CH, CH_3_), 3.81 (d, *J* = 12.7 Hz, 1H, CH), 3.64 (d, *J* = 12.8 Hz, 1H, CH), 3.37 (dt, *J* = 11.1, 5.5 Hz, 1H, CH), 2.92–2.85 (m, 1H, CH), 2.80 (dd, *J* = 17.5, 5.2 Hz, 1H, CH), 2.27–2.17 (m, 1H, CH), 1.99 (s, 3H, CH_3_), 1.57–1.46 (m, 4H, 2CH_2_), 1.29 (t, *J* = 7.1 Hz, 3H, CH_3_), 0.89 (dt, *J* = 9.7, 7.5 Hz, 6H, 2CH_3_). ESI-MS: *m/z* 601.37 [M + H]^+^, C_32_H_41_ClN_2_O_7_ (601.14).

Methyl 4-((4-((((1S,5R,6R)-6-acetamido-3-(ethoxycarbonyl)-5-(pentan-3-yloxy)cyclohex-3-en-1-yl)amino)methyl)phenoxy)methyl)-3-methoxybenzoate (**9b**). Colorless sticky oil, 71% yield. ^1^H NMR (400 MHz, CD_3_OD) δ: 7.64–7.59 (m, 2H, 2Ph-H), 7.50 (d, *J* = 7.7 Hz, 1H, Ph-H), 7.34 (d, *J* = 8.6 Hz, 2H, 2Ph-H), 7.01 (d, *J* = 8.6 Hz, 2H, 2Ph-H), 6.83 (s, 1H, CH), 5.15 (s, 2H, CH_2_), 4.23 (q, *J* = 7.1 Hz, 2H, CH_2_), 4.12 (dd, *J* = 20.2, 7.1 Hz, 2H, CH_2_), 4.08–4.02 (m, 1H, CH), 3.97 (d, *J* = 13.0 Hz, 1H, CH), 3.93 (s, 3H, CH_3_), 3.90 (s, 3H, CH_3_), 3.45–3.38 (m, 1H, CH), 3.30–3.23 (m, 1H, CH), 2.94 (dd, *J* = 17.5, 5.5 Hz, 1H, CH), 2.53–2.40 (m, 1H, CH), 2.02 (s, 3H, CH_3_), 1.59–1.46 (m, 4H, 2CH_2_), 1.30 (t, *J* = 7.1 Hz, 3H, CH_3_), 0.90 (q, *J* = 7.6 Hz, 6H, 2CH_3_). ESI-MS: *m/z* 597.16 [M + H]^+^, C_33_H_44_N_2_O_8_ (596.72).

Methyl 4-((4-((((1S,5R,6R)-6-acetamido-3-(ethoxycarbonyl)-5-(pentan-3-yloxy)cyclohex-3-en-1-yl)amino)methyl)phenoxy)methyl)-2-methoxybenzoate (**10b**). White powder, 67% yield, mp: 133.2–135.3 °C. ^1^H NMR (400 MHz, CD_3_OD) δ: 7.75 (d, *J* = 7.9 Hz, 1H, Ph-H), 7.25 (d, *J* = 8.6 Hz, 2H, 2Ph-H), 7.18 (s, 1H, Ph-H), 7.06 (d, *J* = 7.9 Hz, 1H, Ph-H), 6.96 (d, *J* = 8.6 Hz, 2H, 2Ph-H), 6.77 (s, 1H, CH), 5.13 (s, 2H, CH_2_), 4.21 (q, *J* = 7.1 Hz, 2H, CH_2_), 4.06 (d, *J* = 8.6 Hz, 1H, CH), 4.01–3.86 (m, 5H, CH_2_, CH_3_), 3.86–3.77 (m, 4H, CH, CH_3_), 3.66 (d, *J* = 12.7 Hz, 1H, CH), 3.41–3.34 (m, 1H, CH), 2.92 (d, *J* = 5.5 Hz, 1H, CH), 2.82 (d, *J* = 17.6 Hz, 1H, CH), 2.31–2.19 (m, 1H, CH), 1.99 (s, 3H, CH_3_), 1.58–1.45 (m, 4H, 2CH_2_), 1.29 (t, *J* = 7.1 Hz, 3H, CH_3_), 0.89 (dd, *J* = 17.0, 7.5 Hz, 6H, 2CH_3_). ESI-MS: *m/z* 597.83 [M + H]^+^, C_33_H_44_N_2_O_8_ (596.72).

Ethyl (3R,4R,5S)-4-acetamido-5-((4-((2-nitrobenzyl)oxy)benzyl)amino)-3-(pentan-3-yloxy)cyclohex-1-ene-1-carboxylate (**11b**). White powder, 68% yield, mp: 101.8–103.5 °C. ^1^H NMR (400 MHz, CD_3_OD) δ: 8.12 (d, *J* = 8.2 Hz, 1H, Ph-H), 7.83 (d, *J* = 7.7 Hz, 1H, Ph-H), 7.71 (t, *J* = 7.6 Hz, 1H, Ph-H), 7.55 (t, *J* = 7.8 Hz, 1H, Ph-H), 7.25 (d, *J* = 8.5 Hz, 2H, 2Ph-H), 6.95 (d, *J* = 8.5 Hz, 2H, 2Ph-H), 6.77 (s, 1H, CH), 5.44 (s, 2H, CH_2_), 4.21 (q, *J* = 7.1 Hz, 2H, CH_2_), 4.05 (d, *J* = 8.2 Hz, 1H, CH), 3.90 (dd, *J* = 10.2, 8.7 Hz, 1H, CH), 3.80 (d, *J* = 12.7 Hz, 1H, CH), 3.64 (d, *J* = 12.8 Hz, 1H, CH), 3.41–3.33 (m, 1H, CH), 2.93–2.76 (m, 2H, 2CH), 2.28–2.17 (m, 1H, CH), 1.99 (s, 3H, CH_3_), 1.57–1.43 (m, 4H, 2CH_2_), 1.29 (t, *J* = 7.1 Hz, 3H, CH_3_), 0.89 (dt, *J* = 9.9, 7.4 Hz, 6H, 2CH_3_). ESI-MS: *m/z* 554.5 [M + H]^+^, C_30_H_39_N_3_O_7_ (553.66).

Ethyl (3R,4R,5S)-4-acetamido-5-((4-((3-nitrobenzyl)oxy)benzyl)amino)-3-(pentan-3-yloxy)cyclohex-1-ene-1-carboxylate (**12b**). White powder, 70% yield, mp: 144.2–146.1 °C. ^1^H NMR (400 MHz, CD_3_OD) δ: 8.31 (s, 1H, Ph-H), 8.18 (d, *J* = 8.2 Hz, 1H, Ph-H), 7.84 (d, *J* = 7.7 Hz, 1H, Ph-H), 7.62 (t, *J* = 7.9 Hz, 1H, Ph-H), 7.25 (d, *J* = 8.5 Hz, 2H, 2Ph-H), 6.98 (d, *J* = 8.5 Hz, 2H, 2Ph-H), 6.77 (s, 1H, CH), 5.21 (s, 2H, CH_2_), 4.21 (q, *J* = 7.1 Hz, 2H, CH_2_), 4.05 (d, *J* = 8.2 Hz, 1H, CH), 3.95–3.86 (m, 1H, CH), 3.80 (d, *J* = 12.7 Hz, 1H, CH), 3.64 (d, *J* = 12.7 Hz, 1H, CH), 3.41–3.33 (m, 1H, CH), 2.88 (td, *J* = 9.9, 5.4 Hz, 1H, CH), 2.80 (dd, *J* = 17.5, 5.2 Hz, 1H, CH), 2.22 (ddt, *J* = 15.0, 9.0, 2.4 Hz, 1H, CH), 1.99 (s, 3H, CH_3_), 1.56–1.44 (m, 4H, 2CH_2_), 1.29 (t, *J* = 7.1 Hz, 3H, CH_3_), 0.89 (dd, *J* = 17.2, 7.5 Hz, 6H, 2CH_3_). ESI-MS: *m/z* 554.5 [M + H]^+^, C_30_H_39_N_3_O_7_ (553.66).

Ethyl (3R,4R,5S)-4-acetamido-5-((4-((4-nitrobenzyl)oxy)benzyl)amino)-3-(pentan-3-yloxy)cyclohex-1-ene-1-carboxylate (**13b**). White powder, 71% yield, mp: 118.9–120.1 °C. ^1^H NMR (400 MHz, CD_3_OD) δ: 8.24 (d, *J* = 8.7 Hz, 2H, 2Ph-H), 7.68 (d, *J* = 8.6 Hz, 2H, 2Ph-H), 7.25 (d, *J* = 8.5 Hz, 2H, 2Ph-H), 6.97 (d, *J* = 8.6 Hz, 2H, 2Ph-H), 6.77 (s, 1H, CH), 5.22 (s, 2H, CH_2_), 4.21 (q, *J* = 7.1 Hz, 2H, CH_2_), 4.06 (d, *J* = 8.2 Hz, 1H, CH), 3.90 (dd, *J* = 10.3, 8.6 Hz, 1H, CH), 3.80 (d, *J* = 12.7 Hz, 1H, CH), 3.64 (d, *J* = 12.7 Hz, 1H, CH), 3.41–3.34 (m, 1H, CH), 2.89 (td, *J* = 9.9, 5.4 Hz, 1H, CH), 2.80 (dd, *J* = 17.5, 5.2 Hz, 1H, CH), 2.22 (ddt, *J* = 15.0, 9.4, 2.6 Hz, 1H, CH), 1.99 (s, 3H, CH_3_), 1.57–1.44 (m, 4H, 2CH_2_), 1.29 (t, *J* = 7.1 Hz, 3H, CH_3_), 0.89 (dt, *J* = 9.4, 7.5 Hz, 6H, 2CH_3_). ESI-MS: *m/z* 554.24 [M + H]^+^, C_30_H_39_N_3_O_7_ (553.66).

Ethyl (3R,4R,5S)-4-acetamido-5-((4-((3-carbamoylbenzyl)oxy)benzyl)amino)-3-(pentan-3-yloxy)cyclohex-1-ene-1-carboxylate (**14b**). Light yellow powder, 62% yield, mp: 119.5–121.3 °C. ^1^H NMR (400 MHz, CD_3_OD) δ: 7.89 (s, 1H, Ph-H), 7.75 (d, *J* = 7.8 Hz, 1H, Ph-H), 7.55 (d, *J* = 7.7 Hz, 1H, Ph-H), 7.40 (t, *J* = 7.7 Hz, 1H, Ph-H), 7.23 (d, *J* = 8.6 Hz, 2H, 2Ph-H), 6.95 (d, *J* = 8.6 Hz, 2H, 2Ph-H), 6.74 (s, 1H, CH), 5.08 (s, 2H, CH_2_), 4.16 (q, *J* = 7.1 Hz, 2H, CH_2_), 4.04 (t, *J* = 7.0 Hz, 1H, CH), 4.00–3.88 (m, 2H, 2CH), 3.79 (d, *J* = 12.9 Hz, 1H, CH), 3.33 (dt, *J* = 11.3, 5.6 Hz, 1H, CH), 3.08 (td, *J* = 9.9, 5.4 Hz, 1H, CH), 2.82 (dd, *J* = 17.6, 5.3 Hz, 1H, CH), 2.39–2.26 (m, 1H, CH), 1.94 (s, 3H, CH_3_), 1.51–1.38 (m, 4H, 2CH_2_), 1.23 (t, *J* = 7.1 Hz, 3H, CH_3_), 0.88–0.77 (m, 6H, 2CH_3_). ESI-MS: *m/z* 552.55 [M + H]^+^, C_31_H_41_N_3_O_6_ (551.68).

Ethyl (3R,4R,5S)-4-acetamido-5-((4-((4-carbamoylbenzyl)oxy)benzyl)amino)-3-(pentan-3-yloxy)cyclohex-1-ene-1-carboxylate (**15b**). White powder, 71% yield, mp: 214.5–216.5 °C. ^1^H NMR (400 MHz, CD_3_OD) δ: 7.87 (d, *J* = 8.2 Hz, 2H, 2Ph-H), 7.53 (d, *J* = 8.1 Hz, 2H, 2Ph-H), 7.24 (d, *J* = 8.5 Hz, 2H, 2Ph-H), 6.96 (d, *J* = 8.5 Hz, 2H, 2Ph-H), 6.77 (s, 1H, CH), 5.14 (s, 2H, CH_2_), 4.21 (q, *J* = 7.1 Hz, 2H, CH_2_), 4.05 (d, *J* = 8.1 Hz, 1H, CH), 3.90 (dd, *J* = 10.2, 8.7 Hz, 1H, CH), 3.80 (d, *J* = 12.7 Hz, 1H, CH), 3.64 (d, *J* = 12.7 Hz, 1H, CH), 3.37 (dt, *J* = 11.3, 5.5 Hz, 1H, CH), 2.93–2.85 (m, 1H, CH), 2.81 (dd, *J* = 17.6, 5.0 Hz, 1H, CH), 2.28–2.18 (m, 1H, CH), 1.99 (s, 3H, CH_3_), 1.59–1.43 (m, 4H, 2CH_2_), 1.29 (t, *J* = 7.1 Hz, 3H, CH_3_), 0.89 (dt, *J* = 9.7, 7.4 Hz, 6H, 2CH_3_). ESI-MS: *m/z* 552.17 [M + H]^+^, C_31_H_41_N_3_O_6_ (551.68).

Ethyl (3R,4R,5S)-4-acetamido-3-(pentan-3-yloxy)-5-((4-((3-sulfamoylbenzyl)oxy) benzyl)amino)cyclohex-1-ene-1-carboxylate (**16b**). White powder, 64% yield, mp: 127.5–129.7 °C. ^1^H NMR (400 MHz, CD_3_OD) δ: 7.93 (s, 1H, Ph-H), 7.79 (d, *J* = 7.8 Hz, 1H, Ph-H), 7.60 (d, *J* = 7.7 Hz, 1H, Ph-H), 7.49 (t, *J* = 7.7 Hz, 1H, Ph-H), 7.22 (d, *J* = 8.6 Hz, 2H, 2Ph-H), 6.94 (d, *J* = 8.7 Hz, 2H, 2Ph-H), 6.73 (s, 1H, CH), 5.11 (s, 2H, CH_2_), 4.16 (q, *J* = 7.1 Hz, 2H, CH_2_), 4.02 (d, *J* = 8.1 Hz, 1H, CH), 3.94–3.83 (m, 2H, 2CH), 3.69 (d, *J* = 12.8 Hz, 1H, CH), 3.33 (dt, *J* = 11.3, 5.6 Hz, 1H, CH), 3.00–2.91 (m, 1H, CH), 2.84–2.75 (m, 1H, CH), 2.31–2.20 (m, 1H, CH), 1.94 (s, 3H, CH_3_), 1.52–1.40 (m, 4H, 2CH_2_), 1.23 (t, *J* = 7.1 Hz, 3H, CH_3_), 0.83 (dt, *J* = 9.7, 7.4 Hz, 6H, 2CH_3_). ESI-MS: *m/z* 588.52 [M + H]^+^, C_30_H_41_N_3_O_7_S (587.73).

Ethyl (3R,4R,5S)-4-acetamido-3-(pentan-3-yloxy)-5-((4-((4-sulfamoylbenzyl)oxy) benzyl)amino)cyclohex-1-ene-1-carboxylate (**17b**). White powder, 69% yield, mp: 76.8–78.9 °C. ^1^H NMR (400 MHz, CD_3_OD) δ: 7.83 (d, *J* = 8.3 Hz, 2H, 2Ph-H), 7.54 (d, *J* = 8.3 Hz, 2H, 2Ph-H), 7.28 (d, *J* = 8.6 Hz, 2H, 2Ph-H), 6.97 (d, *J* = 8.6 Hz, 2H, 2Ph-H), 6.77 (s, 1H, CH), 5.14 (s, 2H, CH_2_), 4.17 (q, *J* = 7.1 Hz, 2H, CH_2_), 4.11–3.95 (m, 3H, 3CH), 3.91 (d, *J* = 13.0 Hz, 1H, CH), 3.39–3.32 (m, 1H, CH), 3.23–3.16 (m, 1H, CH), 2.87 (dd, *J* = 17.5, 5.4 Hz, 1H, CH), 2.41 (ddd, *J* = 15.1, 9.7, 2.3 Hz, 1H, CH), 1.95 (s, 3H, CH_3_), 1.53–1.38 (m, 4H, 2CH_2_), 1.24 (t, *J* = 7.1 Hz, 3H, CH_3_), 0.83 (q, *J* = 7.6 Hz, 6H, 2CH_3_). ESI-MS: *m/z* 588.49 [M + H]^+^, C_30_H_41_N_3_O_7_S (587.73).

Ethyl (3R,4R,5S)-4-acetamido-5-((4-((4-(fluorosulfonyl)benzyl)oxy)benzyl)amino) -3-(pentan-3-yloxy)cyclohex-1-ene-1-carboxylate (**18b**). White powder, 72% yield, mp: 109.5–111.3 °C. ^1^H NMR (400 MHz, CD_3_OD) δ: 8.07 (d, *J* = 8.4 Hz, 2H, 2Ph-H), 7.81 (d, *J* = 8.3 Hz, 2H, 2Ph-H), 7.38 (d, *J* = 8.6 Hz, 2H, 2Ph-H), 7.08 (d, *J* = 8.6 Hz, 2H, 2Ph-H), 6.85 (s, 1H, CH), 5.30 (s, 2H, CH_2_), 4.28–4.15 (m, 4H, CH_2_, 2CH), 4.13–4.03 (m, 2H, 2CH), 3.46–3.35 (m, 2H, 2CH), 2.96 (dd, *J* = 17.8, 5.7 Hz, 1H, CH), 2.59–2.48 (m, 1H, CH), 2.02 (s, 3H, CH_3_), 1.53 (tq, *J* = 13.5, 6.8 Hz, 4H, 2CH_2_), 1.30 (t, *J* = 7.1 Hz, 3H, CH_3_), 0.90 (q, *J* = 7.4 Hz, 6H, 2CH_3_). ESI-MS: *m/z* 591.51 [M + H]^+^, C_30_H_39_FN_2_O_7_S (590.71).

#### 4.1.3. General Procedure for the Preparation of Compounds **1c**–**18c**

The intermediates **1b**–**18b** (0.8 mmol) were dissolved in methanol (30 mL) and 1 mol/L aqueous sodium hydroxide (10 mL) was added. The mixed solution was stirred at room temperature for 3–4 h. After completion of the reaction, the methanol was removed under reduced pressure. The residue was taken up in water (30 mL), and the pH value was adjusted to 2–3 with 3 mol/L HCl aqueous solution while the solid was precipitated from the water solution. After that, the solution was filtered and washed with water and dried under vacuum to afford the target compounds **1c**–**18c**.

(3R,4R,5S)-4-acetamido-5-((4-((2-boronobenzyl)oxy)benzyl)amino)-3-(pentan-3-yloxy)cyclohex-1-ene-1-carboxylic acid (**1c**). White powder, 65% yield, mp: 146.3–150.5 °C (along with the decomposition). ^1^H NMR (400 MHz, CD_3_OD) δ: 7.51–7.24 (m, 6H, 6Ph-H), 7.14–7.05 (m, *J* = 9.0 Hz, 2H, 2Ph-H), 6.86 (s, 1H, CH), 5.14 (s, 2H, CH_2_), 4.34 (d, *J* = 12.6 Hz, 1H, CH), 4.27–4.14 (m, 3H, 3CH), 3.62–3.52 (m, 1H, CH), 3.49–3.41 (m, 1H, CH), 3.02 (d, *J* = 16.9 Hz, 1H, CH), 2.70–2.59 (m, 1H, CH), 2.05 (s, 3H, CH_3_), 1.60–1.46 (m, *J* = 19.5, 9.7 Hz, 4H, 2CH_2_), 0.91 (q, *J* = 7.1 Hz, 6H, 2CH_3_). ^13^C NMR (100 MHz, CD_3_OD) δ: 159.5, 139.8, 131.3, 131.2, 131.2, 131.1, 128.2, 126.9, 126.3, 123.12, 115.1, 82.3, 74.5, 69.9, 54.7, 51.5, 25.7, 25.2, 22.0, 21.7, 8.4, 8.12. ESI-MS: *m/z* 525.4 [M + H]^+^, C_28_H_37_BN_2_O_7_ (524.42).

(3R,4R,5S)-4-acetamido-5-((4-((3-boronobenzyl)oxy)benzyl)amino)-3-(pentan-3-yloxy)cyclohex-1-ene-1-carboxylic acid (**2c**). White powder, 53% yield, mp: 209.8–214.3 °C (along with the decomposition). ^1^H NMR (400 MHz, CD_3_OD) δ: 7.73–7.34 (m, 5H, 5Ph-H), 7.08 (t, *J* = 8.6 Hz, 2H, 2Ph-H), 6.97–6.79 (m, 2H, Ph-H, CH), 5.11 (d, *J* = 28.6 Hz, 2H, CH_2_), 4.33 (d, *J* = 13.0 Hz, 1H, CH), 4.27–4.15 (m, *J* = 15.2, 4.9 Hz, 3H, 3CH), 3.55 (td, *J* = 9.6, 5.9 Hz, 1H, CH), 3.46 (dt, *J* = 11.2, 5.5 Hz, 1H, CH), 3.03 (dd, *J* = 17.3, 5.5 Hz, 1H, CH), 2.64 (ddd, *J* = 9.5, 8.5, 4.6 Hz, 1H, CH), 2.06 (s, 3H, CH_3_), 1.55 (tq, *J* = 14.0, 6.8 Hz, 4H, 2CH_2_), 0.92 (q, *J* = 7.4 Hz, 6H, 2CH_3_). ^13^C NMR (100 MHz, CD_3_OD) δ: 173.5, 167.9, 159.8, 138.5, 136.5, 131.1, 130.6, 129.2, 128.1, 127.4, 122.9, 118.1, 115.4, 114.4, 113.8, 82.3, 74.6, 69.5, 54.6, 51.5, 47.1, 26.1, 25.7, 25.2, 22.0, 8.4, 8.2. ESI-MS: *m/z* 523.4 [M − H]^−^, C_28_H_37_BN_2_O_7_ (524.42).

(3R,4R,5S)-4-acetamido-5-((4-((4-boronobenzyl)oxy)benzyl)amino)-3-(pentan-3-yloxy)cyclohex-1-ene-1-carboxylic acid (**3c**). White powder, 63% yield, mp: 168.3–172.1 °C (along with the decomposition). ^1^H NMR (400 MHz, CD_3_OD) δ: 7.70–7.55 (m, 1H, Ph-H), 7.41 (dd, *J* = 8.5, 3.9 Hz, 3H, 3Ph-H), 7.26 (d, *J* = 8.4 Hz, 1H, Ph-H), 7.08 (t, *J* = 8.3 Hz, 2H, 2Ph-H), 6.87 (d, *J* = 8.0 Hz, 1H, CH), 6.79 (d, *J* = 8.5 Hz, 1H, Ph-H), 5.16 (s, 1H, CH), 5.01 (s, 1H, CH), 4.33 (d, *J* = 13.0 Hz, 1H, CH), 4.26–4.15 (m, 3H, 3CH), 3.60–3.50 (m, *J* = 4.2 Hz, 1H, CH), 3.50–3.42 (m, 1H, CH), 3.08–2.97 (m, *J* = 8.5, 7.0 Hz, 1H, CH), 2.71–2.59 (m, 1H, CH), 2.06 (s, 3H, CH_3_), 1.55 (dq, *J* = 14.0, 7.0 Hz, 4H, 2CH_2_), 0.92 (q, *J* = 7.4 Hz, 6H, 2CH_3_). ^13^C NMR (100 MHz, CD_3_OD) δ: 159.9, 159.8, 157.1, 137.8, 133.7, 131.1, 131.1, 129.1, 127.5, 126.2, 124.7, 122.8, 122.6, 115.4, 115.3, 114.8, 82.3, 74.6, 69.7, 69.5, 54.6, 51.5, 29.5, 25.7, 25.2, 22.0, 8.4, 8.2. ESI-MS: *m/z* 523.4 [M − H]^−^, C_28_H_37_BN_2_O_7_ (524.42).

2-((4-((((1S,5R,6R)-6-acetamido-3-carboxy-5-(pentan-3-yloxy)cyclohex-3-en-1-yl)amino)methyl)phenoxy)methyl)benzoic acid (**4c**). White powder, 63% yield, mp: 177.1–178.9 °C. ^1^H NMR (400 MHz, CD_3_OD) δ: 8.06–7.97 (m, 1H, Ph-H), 7.66 (d, *J* = 7.7 Hz, 1H, Ph-H), 7.57–7.50 (m, 1H, Ph-H), 7.48–7.36 (m, 3H, 3Ph-H), 7.05 (d, *J* = 8.7 Hz, 2H, 2Ph-H), 6.85 (s, 1H, CH), 5.50 (s, 2H, CH_2_), 4.31 (d, *J* = 13.0 Hz, 1H, CH), 4.28–4.13 (m, 3H, 3CH), 3.60 (td, *J* = 10.3, 5.6 Hz, 1H, CH), 3.49–3.40 (m, 1H, CH), 3.02 (dd, *J* = 17.3, 5.4 Hz, 1H, CH), 2.71–2.58 (m, 1H, CH), 2.05 (s, 3H, CH_3_), 1.61–1.45 (m, 4H, 2CH_2_), 0.97–0.85 (m, 6H, 2CH_3_). ^13^C NMR (100 MHz, CD_3_OD) δ: 173.5, 169.2, 167.5, 159.8, 138.6, 137.0, 131.9, 131.2, 130.6, 129.2, 127.7, 127.2, 127.2, 122.9, 115.2, 82.3, 74.6, 68.0, 54.6, 51.5, 47.0, 26.0, 25.7, 25.2, 22.0, 8.4, 8.2. HRMS calcd. For C_29_H_36_N_2_O_7_ [M + H]^+^: 525.2595. Found: *m/z* 525.2595.

3-((4-((((1S,5R,6R)-6-acetamido-3-carboxy-5-(pentan-3-yloxy)cyclohex-3-en-1-yl)amino)methyl)phenoxy)methyl)benzoic acid (**5c**). White powder, 70% yield, mp: 146.2–148.5 °C. ^1^H NMR (400 MHz, CD_3_OD) δ: 8.08 (s, 1H, Ph-H), 7.96 (d, *J* = 7.8 Hz, 1H, Ph-H), 7.66 (d, *J* = 7.7 Hz, 1H, Ph-H), 7.52–7.38 (m, 3H, 3Ph-H), 7.08 (d, *J* = 8.6 Hz, 2H, 2Ph-H), 6.84 (s, 1H, CH), 5.19 (s, 2H, CH_2_), 4.32 (d, *J* = 13.0 Hz, 1H, CH), 4.28–4.09 (m, 3H, 3CH), 3.58 (td, *J* = 10.1, 5.6 Hz, 1H, CH), 3.47–3.40 (m, 1H, CH), 3.01 (dd, *J* = 17.3, 5.4 Hz, 1H, CH), 2.72–2.56 (m, 1H, CH), 2.04 (s, 3H, CH_3_), 1.62–1.46 (m, 4H, 2CH_2_), 0.90 (q, *J* = 7.6 Hz, 6H, 2CH_3_). ^13^C NMR (100 MHz, CD_3_OD) δ: 173.5, 168.3, 167.7, 159.6, 137.5, 136.7, 131.5, 131.3, 128.9, 128.3, 127.9, 123.1, 115.4, 82.3, 74.6, 69.0, 54.6, 51.5, 47.1, 26.0, 25.9, 25.2, 22.0, 8.4, 8.2. HRMS calcd for C_29_H_36_N_2_O_7_ [M + H]^+^: 525.2596. Found: *m/z* 525.2595.

4-((4-((((1S,5R,6R)-6-acetamido-3-carboxy-5-(pentan-3-yloxy)cyclohex-3-en-1-yl)amino)methyl)phenoxy)methyl)benzoic acid (**6c**). White powder, 71% yield, mp: 193.3–195.5 °C (along with the decomposition). ^1^H NMR (400 MHz, CD_3_OD) δ: 8.01 (d, *J* = 8.3 Hz, 2H, 2Ph-H), 7.53 (d, *J* = 8.2 Hz, 2H, 2Ph-H), 7.42 (d, *J* = 8.6 Hz, 2H, 2Ph-H), 7.07 (d, *J* = 8.7 Hz, 2H, 2Ph-H), 6.84 (s, 1H, CH), 5.20 (s, 2H, CH_2_), 4.32 (d, *J* = 13.0 Hz, 1H, CH), 4.27–4.13 (m, *J* = 18.8, 8.6 Hz, 3H, 3CH), 3.63–3.54 (m, *J* = 10.1, 5.6 Hz, 1H, CH), 3.48–3.40 (m, 1H, CH), 3.01 (dd, *J* = 17.3, 5.4 Hz, 1H, CH), 2.71–2.58 (m, 1H, CH), 2.04 (s, 3H, CH_3_), 1.60–1.45 (m, 4H, 2CH_2_), 0.90 (q, *J* = 7.6 Hz, 6H, 2CH_3_). ^13^C NMR (100 MHz, CD_3_OD) δ; 173.5, 168.3, 167.6, 159.5, 142.2, 136.7, 131.2, 130.3, 129.6, 127.8, 126.8, 123.1, 115.3, 82.3, 74.6, 68.9, 54.6, 51.5, 47.1, 26.0, 25.7, 25.2, 22.0, 8.4, 8.2. HRMS calcd for C_29_H_36_N_2_O_7_ [M + H]^+^: 525.2596. Found: *m/z* 525.2595.

3-((4-((((1S,5R,6R)-6-acetamido-3-carboxy-5-(pentan-3-yloxy)cyclohex-3-en-1-yl)amino)methyl)phenoxy)methyl)-4-methoxybenzoic acid (**7c**). White powder, 68% yield, mp: 152.1–154.1 °C. ^1^H NMR (400 MHz, CD_3_OD) δ: 8.06–7.95 (m, 2H, 2Ph-H), 7.40 (d, *J* = 8.6 Hz, 2H, 2Ph-H), 7.14–7.02 (m, 3H, 3Ph-H), 6.81 (s, 1H, CH), 5.14 (s, 2H, CH_2_), 4.31 (d, *J* = 13.0 Hz, 1H, CH), 4.24–4.13 (m, *J* = 10.5, 6.4 Hz, 3H, 3CH), 3.94 (s, 3H, CH_3_), 3.57–3.48 (m, 1H, CH), 3.47–3.40 (m, 1H, CH), 3.03 (d, *J* = 5.4 Hz, 1H, CH), 2.71–2.56 (m, *J* = 17.5, 9.9 Hz, 1H, CH), 2.04 (s, 3H, CH_3_), 1.60–1.47 (m, 4H, 2CH_2_), 0.90 (q, *J* = 7.4 Hz, 6H, 2CH_3_). ^13^C NMR (100 MHz, CD_3_OD) δ: 173.4, 168.4, 168.3, 160.7, 159.7, 135.9, 131.3, 131.2, 129.7, 128.6, 125.0, 123.0, 115.2, 109.8, 82.2, 74.6, 64.4, 55.0, 54.6, 51.56, 26.2, 25.7, 25.2, 22.0, 8.4, 8.2. HRMS calcd for C_30_H_38_N_2_O_8_ [M + H]^+^: 555.2703. Found: *m/z* 555.2701.

4-((4-((((1S,5R,6R)-6-acetamido-3-carboxy-5-(pentan-3-yloxy)cyclohex-3-en-1-yl)amino)methyl)phenoxy)methyl)-3-chlorobenzoic acid (**8c**). White powder, 66% yield, mp: 189.3–192.1 °C (along with the decomposition). ^1^H NMR (400 MHz, CD_3_OD) δ: 8.01–7.96 (m, *J* = 3.5 Hz, 1H, Ph-H), 7.89 (dd, *J* = 8.0, 1.3 Hz, 1H, Ph-H), 7.61 (d, *J* = 8.0 Hz, 1H, Ph-H), 7.43 (d, *J* = 8.6 Hz, 2H, 2Ph-H), 7.06 (d, *J* = 8.7 Hz, 2H, 2Ph-H), 6.80 (s, 1H, CH), 5.22 (s, 2H, CH_2_), 4.31 (d, *J* = 13.0 Hz, 1H, CH), 4.24–4.14 (m, 3H, 3CH), 3.56 (td, *J* = 9.9, 5.6 Hz, 1H, CH), 3.47–3.40 (m, 1H, CH), 3.00 (dd, *J* = 17.1, 5.7 Hz, 1H, CH), 2.69–2.58 (m, 1H, CH), 2.04 (s, 3H, CH_3_), 1.59–1.46 (m, 4H, 2CH_2_), 0.90 (q, *J* = 7.4 Hz, 6H, 2CH_3_). ^13^C NMR (100 MHz, CD_3_OD) δ: 173.4, 168.6, 168.2, 159.3, 138.3, 135.7, 134.2, 132.4, 131.2, 130.0, 129.0, 128.5, 127.8, 123.9, 115.2, 82.3, 74.8, 66.7, 54.7, 51.7, 47.1, 26.4, 25.8, 25.2, 22.0, 8.4, 8.2. HRMS calcd for C_29_H_35_ClN_2_O_7_ [M + H]^+^: 559.2210. Found: *m/z* 559.2206.

4-((4-((((1S,5R,6R)-6-acetamido-3-carboxy-5-(pentan-3-yloxy)cyclohex-3-en-1-yl)amino)methyl)phenoxy)methyl)-3-methoxybenzoic acid (**9c**). White powder, 68% yield, mp: 171.2–175.7 °C (along with the decomposition). ^1^H NMR (400 MHz, DMSO-d_6_) δ: 7.98–7.88 (m, 1H, Ph-H), 7.61–7.45 (m, 3H, 2Ph-H, NH), 7.29 (d, *J* = 8.4 Hz, 2H, 2Ph-H), 6.97 (d, *J* = 8.4 Hz, 2H, 2Ph-H), 6.61 (s, 1H, CH), 5.10 (s, 2H, CH_2_), 4.06 (d, *J* = 7.4 Hz, 1H, CH), 3.93–3.72 (m, 6H, CH_3_, 3CH), 3.39–3.32 (m, 1H, CH), 2.92 (dd, *J* = 12.0, 7.0 Hz, 1H, CH), 2.75–2.67 (m, 1H, CH), 2.22 (dd, *J* = 16.6, 10.2 Hz, 1H, CH), 1.87 (s, 3H, CH_3_), 1.50–1.32 (m, 4H, 2CH_2_), 0.87–0.75 (m, 6H, 2CH_3_). ^13^C NMR (100 MHz, DMSO-d_6_) δ: 170.5, 167.9, 167.7, 158.1, 157.0, 137.8, 132.5, 130.4, 130.2, 129.4, 128.8, 122.0, 115.0, 111.5, 81.4, 75.6, 64.8, 56.1, 54.6, 53.6, 48.3, 29.5, 26.1, 25.6, 23.6, 9.9, 9.4. HRMS calcd for C_30_H_38_N_2_O_8_ [M + H]^+^: 555.2706. Found: *m/z* 555.2701.

4-((4-((((1S,5R,6R)-6-acetamido-3-carboxy-5-(pentan-3-yloxy)cyclohex-3-en-1-yl)amino)methyl)phenoxy)methyl)-2-methoxybenzoic acid (**10c**). White powder, 72% yield, mp: 190.5–195.1 °C (along with the decomposition). ^1^H NMR (400 MHz, CD_3_OD) δ: 7.68 (d, *J* = 7.8 Hz, 1H, Ph-H), 7.39 (d, *J* = 8.5 Hz, 2H, 2Ph-H), 7.13 (s, 1H, Ph-H), 7.03 (d, *J* = 8.3 Hz, 3H, 3Ph-H), 6.75 (s, 1H, CH), 5.14 (s, 2H, CH_2_), 4.26 (d, *J* = 13.0 Hz, 1H, CH), 4.21–4.07 (m, *J* = 17.9, 10.8 Hz, 3H, 3CH), 3.85 (s, 3H, CH_3_), 3.51 (dd, *J* = 14.5, 9.1 Hz, 1H, CH), 3.46–3.38 (m, 1H, CH), 2.97 (dd, *J* = 17.3, 5.0 Hz, 1H, CH), 2.61 (dd, *J* = 17.2, 9.7 Hz, 1H, CH), 2.03 (s, 3H, CH_3_), 1.61–1.42 (m, 4H, 2CH_2_), 0.89 (dd, *J* = 13.5, 7.2 Hz, 6H, 2CH_3_). ^13^C NMR (100 MHz, CD_3_OD) δ: 173.3, 169.8, 169.4, 159.4, 158.5, 142.3, 134.9, 131.0, 130.9, 129.9, 123.8, 122.0, 118.6, 115.3, 110.4, 82.2, 74.9, 69.0, 55.0, 54.6, 51.8, 48.1, 26.7, 25.8, 25.2, 22.0, 8.4, 8.2. HRMS calcd for C_30_H_38_N_2_O_8_ [M + H]^+^: 555.2705. Found: *m/z* 555.2701.

(3R,4R,5S)-4-acetamido-5-((4-((2-nitrobenzyl)oxy)benzyl)amino)-3-(pentan-3-yloxy)cyclohex-1-ene-1-carboxylic acid (**11c**). White powder, 68% yield, mp: 150.8–152.7 °C. ^1^H NMR (400 MHz, CD_3_OD) δ: 8.00 (d, *J* = 8.1 Hz, 1H, Ph-H), 7.70 (d, *J* = 7.7 Hz, 1H, Ph-H), 7.60 (t, *J* = 7.5 Hz, 1H, Ph-H), 7.45 (t, *J* = 7.5 Hz, 1H, Ph-H), 7.31 (d, *J* = 8.4 Hz, 2H, 2Ph-H), 6.95 (d, *J* = 8.4 Hz, 2H, 2Ph-H), 6.64 (s, 1H, CH), 5.36 (s, 2H, CH_2_), 4.17 (d, *J* = 12.9 Hz, 1H, CH), 4.11–3.98 (m, 3H, 3CH), 3.41–3.28 (m, 2H, 2CH), 2.86 (dd, *J* = 17.1, 4.3 Hz, 1H, CH), 2.49 (dd, *J* = 17.2, 9.2 Hz, 1H, CH), 1.92 (s, 3H, CH_3_), 1.48–1.37 (m, 4H, 2CH_2_), 0.79 (dd, *J* = 13.0, 7.3 Hz, 6H, 2CH_3_). ^13^C NMR (100 MHz, CD_3_OD) δ: 173.3, 169.5, 159.2, 147.8, 134.6, 133.4, 132.6, 131.1, 130.0, 128.9, 128.7, 124.5, 124.4, 115.2, 82.1, 74.7, 66.8, 54.7, 51.9, 48.1, 26.7, 25.8, 25.2, 21.9, 8.4, 8.2. HRMS calcd for C_28_H_35_N_3_O_7_ [M + H]^+^: 526.2551. Found: *m/z* 526.2548.

(3R,4R,5S)-4-acetamido-5-((4-((3-nitrobenzyl)oxy)benzyl)amino)-3-(pentan-3-yloxy)cyclohex-1-ene-1-carboxylic acid (**12c**). White powder, 79% yield, mp: 152.5–155.1 °C. ^1^H NMR (400 MHz, CD_3_OD) δ: 8.33 (s, 1H, Ph-H), 8.20 (d, *J* = 8.2 Hz, 1H, Ph-H), 7.87 (d, *J* = 7.6 Hz, 1H, Ph-H), 7.65 (t, *J* = 7.9 Hz, 1H, Ph-H), 7.45 (d, *J* = 8.5 Hz, 2H, 2Ph-H), 7.12 (d, *J* = 8.5 Hz, 2H, 2Ph-H), 6.80 (s, 1H, CH), 5.28 (s, 2H, CH_2_), 4.31 (d, *J* = 13.0 Hz, 1H, CH), 4.17 (dd, *J* = 10.4, 7.2 Hz, 3H, 3CH), 3.53 (dd, *J* = 14.8, 9.4 Hz, 1H, CH), 3.49–3.42 (m, 1H, CH), 3.00 (dd, *J* = 16.2, 6.1 Hz, 1H, CH), 2.63 (dd, *J* = 17.4, 9.5 Hz, 1H, CH), 2.06 (s, 3H, CH_3_), 1.61–1.48 (m, 4H, 2CH_2_), 0.92 (dd, *J* = 13.2, 7.3 Hz, 6H, 2CH_3_). ^13^C NMR (100 MHz, CD_3_OD) δ: 173.4, 169.0, 159.2, 148.4, 139.5, 135.1, 133.1, 131.2, 129.5, 123.9, 122.4, 121.6, 115.3, 82.2, 74.7, 68.2, 54.7, 51.7, 47.1, 26.5, 25.7, 25.2, 22.0, 8.4, 8.2. HRMS calcd for C_28_H_35_N_3_O_7_ [M + H]^+^: 526.2550. Found: *m/z* 526.2548.

(3R,4R,5S)-4-acetamido-5-((4-((4-nitrobenzyl)oxy)benzyl)amino)-3-(pentan-3-yloxy)cyclohex-1-ene-1-carboxylic acid (**13c**). khaki powder, 68% yield, mp: 184.1–187.5 °C (along with the decomposition). ^1^H NMR (400 MHz, CD_3_OD) δ: 8.26 (d, *J* = 8.7 Hz, 2H, 2Ph-H), 7.71 (d, *J* = 8.6 Hz, 2H, 2Ph-H), 7.46 (d, *J* = 8.6 Hz, 2H, 2Ph-H), 7.12 (d, *J* = 8.6 Hz, 2H, 2Ph-H), 6.87 (s, 1H, CH), 5.29 (s, 2H, CH_2_), 4.35 (d, *J* = 13.0 Hz, 1H, CH), 4.29–4.15 (m, 3H, 3CH), 3.62 (td, *J* = 10.0, 5.6 Hz, 1H, CH), 3.50–3.43 (m, 1H, CH), 3.03 (dd, *J* = 17.4, 5.3 Hz, 1H, CH), 2.72–2.60 (m, 1H, CH), 2.07 (s, 3H, CH_3_), 1.62–1.48 (m, 4H, 2CH_2_), 0.92 (q, *J* = 7.3 Hz, 6H, 2CH_3_). ^13^C NMR (100 MHz, CD_3_OD) δ: 173.5, 167.6, 159.3, 147.6, 144.6, 136.8, 131.3, 127.8, 127.7, 123.4, 123.3, 115.3, 82.3, 74.6, 68.3, 54.6, 51.5, 26.0, 25.7, 25.2, 22.0, 8.4, 8.2. HRMS calcd for C_28_H_35_N_3_O_7_ [M + H]^+^: 526.2549. Found: *m/z* 526.2548.

(3R,4R,5S)-4-acetamido-5-((4-((3-carbamoylbenzyl)oxy)benzyl)amino)-3-(pentan-3-yloxy)cyclohex-1-ene-1-carboxylic acid (**14c**). White powder, 62% yield, mp: 138.2–140.5 °C. ^1^H NMR (400 MHz, CD_3_OD) δ: 7.98 (s, 1H, Ph-H), 7.84 (d, *J* = 7.7 Hz, 1H, Ph-H), 7.64 (d, *J* = 7.5 Hz, 1H, Ph-H), 7.57–7.38 (m, 3H, 3Ph-H), 7.11 (d, *J* = 8.5 Hz, 2H, 2Ph-H), 6.88 (s, 1H, CH), 5.20 (s, 2H, CH_2_), 4.35 (d, *J* = 13.0 Hz, 1H, CH), 4.22 (dt, *J* = 23.4, 9.4 Hz, 3H, 3CH), 3.60 (td, *J* = 10.0, 6.1 Hz, 1H, CH), 3.51–3.43 (m, 1H, CH), 3.04 (dd, *J* = 17.3, 4.7 Hz, 1H, CH), 2.66 (dd, *J* = 17.1, 9.6 Hz, 1H, CH), 2.07 (s, 3H, CH_3_), 1.63–1.47 (m, 4H, 2CH_2_), 1.00–0.84 (m, 6H, 2CH_3_). ^13^C NMR (100 MHz, CD_3_OD) δ: 173.5, 170.6, 159.6, 137.6, 136.9, 133.9, 131.2, 130.5, 128.4, 126.8, 126.4, 124.7, 123.0, 115.4, 82.3, 74.5, 69.2, 67.5, 54.7, 51.5, 29.5, 25.7, 25.2, 22.0, 8.4, 8.1. HRMS calcd for C_29_H_37_N_3_O_6_ [M + H]^+^: 524.2758. Found: *m/z* 524.2755.

(3R,4R,5S)-4-acetamido-5-((4-((4-carbamoylbenzyl)oxy)benzyl)amino)-3-(pentan-3-yloxy)cyclohex-1-ene-1-carboxylic acid (**15c**). White powder, 73% yield, mp: 172.5–174.7 °C. ^1^H NMR (400 MHz, CD_3_OD) δ: 7.87 (d, *J* = 8.2 Hz, 2H, 2Ph-H), 7.52 (d, *J* = 8.2 Hz, 2H, 2Ph-H), 7.41 (d, *J* = 8.6 Hz, 2H, 2Ph-H), 7.07 (d, *J* = 8.6 Hz, 2H, 2Ph-H), 6.82 (s, 1H, CH), 5.19 (s, 2H, CH_2_), 4.31 (d, *J* = 13.0 Hz, 1H, CH), 4.26–4.13 (m, 3H, 3CH), 3.57 (td, *J* = 10.0, 5.7 Hz, 1H, CH), 3.44 (p, *J* = 5.6 Hz, 1H, CH), 3.00 (dd, *J* = 17.8, 4.9 Hz, 1H, CH), 2.63 (dd, *J* = 17.4, 9.8 Hz, 1H, CH), 2.04 (s, 3H, CH_3_), 1.59–1.46 (m, 4H, 2CH_2_), 0.90 (q, *J* = 7.3 Hz, 6H, 2CH_3_). ^13^C NMR (100 MHz, CD_3_OD) δ: 173.4, 170.5, 168.2, 159.5, 141.0, 136.2, 133.1, 131.2, 128.4, 127.5, 126.4, 123.3, 115.3, 82.3, 74.6, 68.9, 54.7, 51.6, 47.9, 26.2, 25.7, 25.2, 22.0, 8.4, 8.2. HRMS calcd for C_29_H_37_N_3_O_6_ [M + H]^+^: 524.2758. Found: *m/z* 524.2755.

(3R,4R,5S)-4-acetamido-3-(pentan-3-yloxy)-5-((4-((3-sulfamoylbenzyl)oxy) benzyl)amino)cyclohex-1-ene-1-carboxylic acid (**16c**). White powder, 70% yield, mp: 193.2–198.1 °C (along with the decomposition). ^1^H NMR (400 MHz, DMSO-*d_6_*) δ: 7.84 (d, *J* = 8.2 Hz, 2H, 2Ph-H), 7.61 (d, *J* = 8.3 Hz, 2H, 2Ph-H), 7.35–7.26 (m, 2H, 2Ph-H), 6.97 (d, *J* = 8.6 Hz, 2H, 2Ph-H), 6.61 (s, 1H, CH), 5.19 (s, 2H, CH_2_), 4.07 (d, *J* = 7.4 Hz, 1H, CH), 3.88–3.66 (m, *J* = 18.9, 13.8 Hz, 3H, 3CH), 3.40–3.32 (m, 1H, CH), 2.90 (d, *J* = 7.7 Hz, 1H, CH), 2.75–2.65 (m, *J* = 15.4, 4.7 Hz, 1H, CH), 2.27–2.12 (m, 1H, CH), 1.86 (s, 3H, CH_3_), 1.42 (qd, *J* = 14.0, 7.2 Hz, 4H, 2CH_2_), 0.87–0.74 (m, 6H, 2CH_3_). ^13^C NMR (100 MHz, DMSO-*d_6_*) δ: 170.3, 167.8, 157.7, 143.9, 143.8, 141.7, 137.9, 130.2, 129.3, 128.1, 126.3, 115.1, 81.4, 75.6, 68.8, 65.5, 54.5, 53.6, 30.9, 26.0, 25.6, 23.6, 9.9, 9.3. HRMS calcd for C_28_H_37_N_3_O_7_S [M + H]^+^: 560.2427. Found: *m/z* 560.2425.

(3R,4R,5S)-4-acetamido-3-(pentan-3-yloxy)-5-((4-((4-sulfamoylbenzyl)oxy) benzyl)amino)cyclohex-1-ene-1-carboxylic acid (**17c**). White powder, 69% yield, mp: 158.2–160.5 °C. ^1^H NMR (400 MHz, CD_3_OD) δ: 7.99 (s, 1H, Ph-H), 7.85 (d, *J* = 7.6 Hz, 1H, Ph-H), 7.66 (d, *J* = 7.5 Hz, 1H, Ph-H), 7.55 (t, *J* = 7.7 Hz, 1H, Ph-H), 7.42 (d, *J* = 8.4 Hz, 2H, 2Ph-H), 7.09 (d, *J* = 8.4 Hz, 2H, 2Ph-H), 6.83 (s, 1H, CH), 5.21 (s, 2H, CH_2_), 4.31 (d, *J* = 13.1 Hz, 1H, CH), 4.25–4.12 (m, 3H, 3CH), 3.53 (td, *J* = 9.7, 6.2 Hz, 1H, CH), 3.48–3.41 (m, 1H, CH), 3.00 (dd, *J* = 17.4, 5.1 Hz, 1H, CH), 2.62 (dd, *J* = 17.4, 9.6 Hz, 1H, CH), 2.04 (s, 3H, CH_3_), 1.60–1.46 (m, 4H, 2CH_2_), 0.90 (q, *J* = 7.1 Hz, 6H, 2CH_3_). ^13^C NMR (100 MHz, CD_3_OD) δ: 173.4, 168.0, 159.5, 144.1, 138.4, 136.1, 131.1, 130.6, 128.9, 128.3, 125.2, 124.5, 123.5, 115.4, 82.3, 74.5, 68.8, 54.6, 51.6, 47.9, 26.1, 25.7, 25.2, 21.9, 8.4, 8.1. HRMS calcd for C_28_H_37_N_3_O_7_S [M + H]^+^: 560.2428. Found: *m/z* 560.2425.

(3R,4R,5S)-4-acetamido-3-(pentan-3-yloxy)-5-((4-((4-sulfobenzyl)oxy)benzyl) amino)cyclohex-1-ene-1-carboxylic acid (**18c**). Off-white powder, 58% yield, mp: 244.5–248.7 °C (along with the decomposition). ^1^H NMR (400 MHz, DMSO-*d_6_*) δ: 12.55 (s, 1H, COOH), 8.77 (d, *J* = 165.8 Hz, 1H, NH), 8.01 (d, *J* = 9.1 Hz, 1H, NH), 7.60 (d, *J* = 8.1 Hz, 2H, 2Ph-H), 7.45–7.30 (m, 4H, 4Ph-H), 7.06 (d, *J* = 8.6 Hz, 2H, 2Ph-H), 6.64 (s, 1H, CH), 5.18 (s, 2H, CH_2_), 4.23–4.13 (m, 2H, overlapped, 2CH), 4.08 (d, *J* = 13.1 Hz, 1H, CH), 3.99 (dd, *J* = 20.0, 8.9 Hz, 1H, CH), 3.31–3.20 (m, 2H, 2CH), 2.83 (dd, *J* = 17.0, 4.9 Hz, 1H, CH), 2.58 (dd, *J* = 16.5, 11.0 Hz, 1H, CH), 1.91 (s, 3H, CH_3_), 1.51–1.34 (m, 4H, 2CH_2_), 0.81 (dt, *J* = 17.9, 7.3 Hz, 6H, 2CH_3_). ^13^C NMR (100 MHz, DMSO-*d_6_*) δ: 171.3, 167.3, 158.9, 148.2, 137.9, 137.6, 131.9, 127.98, 127.4, 126.1, 115.7, 81.7, 75.0, 69.2, 54.5, 51.2, 46.2, 26.2, 26.0, 25.5, 23.9, 9.8, 9.4. HRMS calcd for C_28_H_36_N_2_O_8_S [M + H]^+^: 561.2267_._ Found: *m/z* 561.2265

### 4.2. Neuraminidase Enzyme Inhibitory Assay In Vitro

The influenza neuraminidase activity was measured according to the standard method [27,28,29,35]. The NAs (A/PuertoRico/8/1934 (H1N1), A/Babol/36/2005 (H3N2) and A/Anhui/1/2005 (H5N1-H274Y)) were purchased from Sino Biological Inc., and influenza virus suspensions of A/Goose/Guangdong/SH7/2013 (H5N1) and/Goose/Jiangsu/1306/2014 (H5N8) were harvested from the allantoic fluid of influenza virus-infected chicken embryo layer. The substrate, named 2’-(4-methylumbelliferyl)-*α*-D-acetylneuraminic acid sodium salt hydrate (4-MUNANA) (Sigma (St. Louis, MO, USA); M8639), was cleaved by NA to yield a quantifiable fluorescent product. The compounds were first dissolved in DMSO, and then diluted to the corresponding concentrations with MES buffer (1.27 g 2-(N-morpholino)-ethanesulfonic acid and 0.09 g CaCl_2_ in 200 mL Milli-Q water). To a 96-well fluorescent plate, 10 μL of diluted virus supernatant or NA assay diluent, 70 μL of MES buffer and 10 μL of test compounds at different concentrations were added successively. After incubation for 10 min at 37 °C, 10 μL of substrate were added to each well to start the reaction and the plate was further incubated for 40 min. The reaction was stopped by the addition of 150 μL of termination solution (3 g glycine (40 mmol) and 1.6 g NaOH (40 mmol) in 200 mL Milli-Q water). Fluorescence was measured (Ex = 365 nm and Em = 460 nm) with a microplate reader (Molecular Devices; SpectraMax iD5), and substrate blanks were subtracted from the sample readings. The values of IC_50_ (50%-inhibitory concentration) were determined from the dose–response curves by plotting the percentage inhibition of NA activity versus the concentration of the compounds.

### 4.3. In Vitro Anti-Influenza Virus Assay and Cytotoxicity Assay in Chicken Embryo Fibroblast (CEF)

The anti-influenza activity (EC_50_) and cytotoxicity (CC_50_) of the novel synthesized compounds were evaluated with H5N1 and H5N8 strains in Chicken Embryo Fibroblast cells (CEF) using Cell Counting Kit-8 (CCK-8; Dojindo Laboratories) method as previously described [28,29,35]. The tested compounds and positive control were dissolved in DMSO in advance and diluted in assay media (1% FBS in DMEM). Aliquots of 50 μL of influenza virus suspension (H5N1, H5N8) were mixed with equal volumes of solutions of the tested compounds. The mixtures were added to CEFs in 96-well plates and incubated at 37 °C under 5% CO_2_ for 48 h. After incubation, 100 μL per well of CCK-8 reagent solution (10 μL of CCK-8 in 90 μL of media) was added according to the manufacturer’s manual. Incubating at 37 °C for 90 min, the absorbance at 450 nm was read on a microplate reader. The EC_50_ values were calculated by fitting the curve of percent CPE (cytopathic effect) versus inhibitor concentration. The values of CC_50_ (50% cytotoxic concentration) of the compounds to CEFs were determined in the same manner as EC_50_ but without virus infection.

### 4.4. Plaque Reduction Assay (PRA) in MDCK Cells

The antiviral activity of selected compounds against the H1N1 and H3N2 viruses was investigated by PRA, as previously described [36,37,38,39,40,41,42]. Briefly, MDCK cells were seeded at 5 × 10^5^ cells/well into 12-well plates, and then incubated at 37 °C for 24 h. The following day, the culture medium was removed and the monolayers were first washed with serum-free DMEM and then infected with FluA virus (PR8 or WSN strain, 40 PFU/well) in DMEM supplemented with 1 mg/mL of TPCK-treated trypsin (Worthington Biochemical Corporation) and 0.14% BSA in the presence of different concentrations of test compounds for 1 h at 37 °C. After virus adsorption, serum-free medium containing 1 mg/mL of TPCK-treated trypsin, 0.14% BSA, 1.2% Avicel, and test compounds at the same concentrations was then added to cells. Oseltamivir carboxylic acid (**OSC**) and zanamivir (**ZAN**) were included in each of the experiments as the reference compounds. At 48 h post-infection, cells’ monolayers were fixed with 4% formaldehyde and stained with 0.1% toluidine blue. The viral plaques were counted, and the mean plaque number in the DMSO-treated control was set at 100%.

### 4.5. Cytotoxicity Assay in MDCK Cells

Cytotoxicity of representative compounds was assessed in MDCK cells by the 3-(4,5-dimethylthiazol-2-yl)-2,5-diphenyltetrazolium bromide (MTT) method as previously reported [36,37,38,39,40,41,42]. MDCK cells (seeded at density of 2 × 10^4^ cells per well) were cultured in 96-well plates for 24 h and then treated with serial dilutions of test compounds, or the same volume of DMSO as a control, in DMEM supplemented with 10% FBS. After 48 h of incubation at 37 °C, MTT solution (5 mg/mL in PBS) was added to each well and plates were incubated for further 4 h at 37 °C in a CO_2_ incubator. Successively, a solubilization solution (10% SDS, 0.01 N HCl) was added to lyse cells and incubated for 3 h at 37 °C. Finally, absorbance was read at the wavelength of 620 nm on a microtiter plate reader (Tecan Sunrise). Values obtained from the wells treated with no compound were set as 100% of viable cells.

### 4.6. Molecular Docking

The docking simulation studies of compound **2c** were performed using Schrödinger Maestro 11.8 software. Then, **2c** was drawn using ChemDraw professional software, imported to Maestro and then optimized using the LigPrep module. LigPrep performed the 3D structure conversion from 2D with accurate chiralities and an OPLS force field was applied to generate stable conformer with minimum potential energy [43]. The crystal structures of N1 (H5N1, PDB code: 2HU0) and N1-H274Y (H5N1-H274Y, PDB code: 3CL0) were retrieved from molecular dynamics (MD) simulations using Amber14 software as previously described [28]. Before docking, the protein structures were prepared using the “Protein Preparation Wizard” tool of Maestro 11.8 by the addition of hydrogen atoms, and the removal of unwanted water molecules [44]. Furthermore, the grid was generated in the protein using the receptor grid generation tool of the Glide module. Molecular docking of **2c** was conducted using standard Glide protocol (Schrödinger Maestro 11.8) [45]. Docking results were analyzed according to Glide score or Glide G-score scoring function and visualized using the software of PyMOL version 1.5.

### 4.7. Acute Toxicity Experiment

Kunming mice (18–22 g and 4–5 weeks old) were purchased from the Animal Experimental Center of Shandong University. The research protocol complied strictly in accordance with the institutional guidelines of Animal Care and Use Committee (AEWC) at Shandong University. Animals were fed at 25 ± 1 °C, and relative humidity was 60 ± 10%, 12 h of light and 12 h of darkness for every day and the mice were given free access to food and water. To evaluate the acute toxicity of **2c** in mice, we divided 20 healthy Kunming mice into two groups (5 female mice and 5 male mice per group). Compound **2c** was suspended in a mixture of 5% DMSO, 20% PEG-400 and 75% water at a concentrations of 0.1 g/mL, administered intra-gastrically by gavage after the mice had fasted for 12 h. A dose of 1 g/kg of **2c** was administered to 10 mice (5 males and 5 females), while the mice in control groups (another 5 male and 5 female mice) were given the same volume of the vehicle solution without **2c**. Death, abnormal behaviors and body weight were monitored every day for one week. At the end of the experiment, all animals were sacrificed for subsequent experimental studies [28,29,35].

## Data Availability

Not applicable.

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
