# Peer review of "Discovery of Novel Boron-Containing *N*-Substituted Oseltamivir Derivatives as Anti-Influenza A Virus Agents for Overcoming N1-H274Y Oseltamivir-Resistant"

_molecules, 2022, doi:10.3390/molecules27196426_

Round 1
Reviewer 1 Report
In this manuscript, Jia et al. synthesized 18 oseltamivir derivatives and found two boronic acid derivatives that interestingly showed 2 and 4-fold increases compared to OC in the in vitro activity against H5N1-H274Y NA. However, no H5N1-H274Y virus was used to test the antiviral activity. The authors need to test against the H5N1-H274Y strain because the two boronic acid derivatives did not show improved activity in either the enzymatic assay or antiviral experiments against OC-susceptible strains.
Other minor issues include a) the compound numbers that need to be emboldened, b) compound formulas (CD3OD, etc.), and scientific terms (IC50, etc.) that need to have numbers in subscript, c) scientific notation, e.g. 0.0011 is supposed to be recorded as 1.1 x 10^-3, etc.
Reviewer 2 Report
In the manuscript submitted, the authors present the design and synthesis of boron-containing N-substituted oseltamivir derivatives, as well as their biological evaluation as anti-influenza A virus agents.
Overall, the authors have provided a comprehensive and well-organized manuscript that presents interesting results.
However, some issues should be considered:
¾ The title of the article is cut out in the main document, and the authors should consider making it more concise.
¾ Because of the importance of ADME parameters of designed compounds, the authors should present their predicted physicochemical properties. In addition, it would be of great interest to determine experimentally the lipophilicity of the compounds tested.
I consider this article suitable for publication in Molecules, after minor corrections.
Round 2
Reviewer 1 Report
Please note that "10^-3" means 10 to the power of -3. The authors do not need to add "^" if -3 is given in superscript.